# SRSF10 is essential for progenitor spermatogonia expansion by regulating alternative splicing

Wenbo Liu[1,2]*[†], Xukun Lu[3†], Zheng-Hui Zhao[4†], Ruibao SU[5†], Qian-Nan Li Li[4], Yue Xue[4], Zheng Gao[1,2], Si-Min Sun Sun[4], Wen-Long Lei[4], Lei Li[1,2], Geng An[1,2], Hanyan Liu[1,2], Zhiming Han[4], Ying-Chun Ouyang[4], Yi Hou[4], Zhen-Bo Wang[4]*, Qing-Yuan Sun[5]*, Jianqiao Liu[1,2]*

[1]Department of Obstetrics and Gynecology, Center for Reproductive Medicine, Guangdong Provincial Key Laboratory of Major Obstetric Diseases, The Third Affiliated Hospital of Guangzhou Medical University, Guangzhou, China; [2]Key Laboratory for Reproductive Medicine of Guangdong Province, The Third Affiliated Hospital of Guangzhou Medical University, Guangzhou, China; [3]Center for Stem Cell Biology and Regenerative Medicine, MOE Key Laboratory of Bioinformatics, School of Life Sciences, Tsinghua University, Beijing, China; [4]State Key Laboratory of Stem Cell and Reproductive Biology, Institute of Zoology, Chinese Academy of Sciences, Beijing, China; [5]Fertility Preservation Lab, Guangdong-Hong Kong Metabolism & Reproduction Joint Laboratory, Reproductive Medicine Center, Guangdong Second Provincial General Hospital, Guangzhou, China

*For correspondence:
liuwenbo@gzhmu.edu.cn (WL);
wangzb@ioz.ac.cn (Z-BoW);
sunqy@gd2h.org.cn (Q-YuanS);
liujqssz@gzhmu.edu.cn (JL)

[†]These authors contributed equally to this work

Competing interest: The authors declare that no competing interests exist.

**Abstract** Alternative splicing expands the transcriptome and proteome complexity and plays essential roles in tissue development and human diseases. However, how alternative splicing regulates spermatogenesis remains largely unknown. Here, using a germ cell-specific knockout mouse model, we demonstrated that the splicing factor *Srsf10* is essential for spermatogenesis and male fertility. In the absence of SRSF10, spermatogonial stem cells can be formed, but the expansion of Promyelocytic Leukemia Zinc Finger (PLZF)-positive undifferentiated progenitors was impaired, followed by the failure of spermatogonia differentiation (marked by KIT expression) and meiosis initiation. This was further evidenced by the decreased expression of progenitor cell markers in bulk RNA-seq, and much less progenitor and differentiating spermatogonia in single-cell RNA-seq data. Notably, SRSF10 directly binds thousands of genes in isolated THY[+] spermatogonia, and *Srsf10* depletion disturbed the alternative splicing of genes that are preferentially associated with germ cell development, cell cycle, and chromosome segregation, including *Nasp*, *Bclaf1*, *Rif1*, *Dazl*, *Kit*, *Ret*, and *Sycp1*. These data suggest that SRSF10 is critical for the expansion of undifferentiated progenitors by regulating alternative splicing, expanding our understanding of the mechanism underlying spermatogenesis.

## Editor's evaluation

The overall conclusion that SRSF10 plays important roles during mouse spermatogenesis, especially during the proliferation of spermatogonial stem cells and meiosis, is supported by the results. Furthermore, the RNA sequencing data could be an excellent resource for other investigators interested in further exploration of the regulation of spermatogenesis by alternative RNA splicing. This manuscript represents a valuable contribution to the literature on the role of alternative splicing in spermatogenesis. It should be of considerable interest to investigators

studying post-transcriptional regulation of spermatogenesis, oogenesis, and animal development in general.

## Introduction

Spermatogenesis is a complex and highly coordinated process during which spermatogonial stem cells (SSCs) give rise to haploid spermatozoa sustainably throughout life. The balance of self-renewal and differentiation of SSCs is fundamental for the maintenance of spermatogenesis throughout life. Over-self-renewal of SSCs will lead to stem cell accumulation and impair spermatogenesis, and even induce tumor. Conversely, overdifferentiation will lead to SSC exhaustion, and thus progressive loss of germ cells and Sertoli cell-only syndrome (*de Rooij, 2001*). Around embryonic day 15 (E15) in mice, male germ cells do not enter meiosis but are arrested at the G0/G1 phase and are referred to as prospermatogonia. The prospermatogonia resume mitotic proliferation after birth and migrate from the center to the peripheral basement membrane, entering the appropriate environment (stem cell niche) to develop into SSCs. SSCs can self-renew to sustain the stem cell pool or differentiate to generate progenitor cells destined for differentiation (*Oatley and Brinster, 2006*; *de Rooij, 2017*). The stem cells and progenitor spermatogonia are collectively called undifferentiated spermatogonia which include A-single (As, single cells), A-paired (Apr, 2 cells interconnected by cytoplasmic bridges), and A-aligned (Aal, 4, 8, or 16 cells interconnected by cytoplasmic bridges) spermatogonia. Then, Aal spermatogonia transform into type A1 spermatogonia and further differentiate into A2, A3, A4, intermediate (In), and B spermatogonia. All these cell types are identified as differentiating spermatogonia (*Song and Wilkinson, 2014*). Then, type B spermatogonia will divide into pre-leptotene spermatocytes which are the last stage of the mitotic phase and have the competence to enter meiosis. After two rounds of meiotic divisions, haploid spermatids are formed (*Subash and Kumar, 2021*). A large number of genes and multiple regulatory layers of gene expression, including transcriptional and post-transcriptional regulation are reported to be involved in such a complex process to ensure successful spermatogenesis. However, much remains to be understood regarding the homeostasis of SSCs in this intricate process.

Alternative splicing (AS) is a very important and universal post-transcriptional regulatory mechanism to expand the diversity of transcripts and proteins from a limited number of genes (*Maniatis and Tasic, 2002*). Importantly, AS occurs more frequently in higher mammals (90–95% of human genes) and complex organs (brain, heart, and testes; *Wang et al., 2008*; *Merkin et al., 2012*), suggesting that AS contributes to the complexity of the organism. Large-scale analysis based on expressed sequence tags (ESTs) and deep sequencing reveals that AS is at an unusually high level in testes, where the transcription of the genome is substantially more widespread than in other organs (*Yeo et al., 2004*; *Soumillon et al., 2013*; *Ramsköld et al., 2009*), indicating the regulatory functions of AS in the development of testes. Recently, single-cell RNA sequencing (scRNA-seq) reveals that the gene ontology (GO) terms of mRNA splicing and mRNA processing are enriched in the spermatogonia (*Green et al., 2018*). Moreover, many RNA splicing proteins are more highly expressed in type A spermatogonia and in pachytene spermatocyte cell clusters (*Schmid et al., 2013*; *Gan et al., 2013*). AS isoform regulation impacts greatly on the germ cell transcriptome as cells transit from the mitotic to meiotic stages of spermatogenesis (*Hannigan et al., 2017*), suggesting that AS may be very effective for the mitotic division of spermatogonia. Recently, the RNA helicase DDX5 is shown to play essential post-transcriptional roles in the maintenance and function of spermatogonia via regulating the splicing of functional genes in spermatogonia (*Legrand et al., 2019*). *Mettl3*-mediated m6A regulates the AS of genes functioning in spermatogenesis and is critical for spermatogonial differentiation and meiosis initiation (*Xu et al., 2017*). Our previous study has also revealed that *Bcas2* regulates AS in spermatogonia and is involved in meiosis initiation (*Liu et al., 2017*). Despite these encouraging findings, our knowledge of the principle and mechanistic regulation of spermatogenesis by AS, such as why mRNA splicing and processing are robust in spermatogonia, how the alternatively spliced mRNAs are controlled developmentally, and what the roles of different RNA-binding proteins (RBPs) are in this complicated process is still very limited.

SRSF10 is an atypical SR protein, and it contains an RNA recognition motif in the N-terminus and arginine/serine amino acid sequences (RS domain) in the C-terminus (*Shin and Manley, 2002*). Numerous in vitro and in vivo experiments demonstrate that the phosphorylation status of SRSF10 is

importance for its activity (*Shin and Manley, 2002*; *Shin et al., 2005*; *Shin et al., 2004*; *Feng et al., 2008*), and SRSF10 is critical for many physiological processes by regulating the correct functional AS (*Feng et al., 2009*; *Li et al., 2014*; *Zhou et al., 2014b*; *Wei et al., 2015*). SRSF10 is highly expressed in neural and testes tissues, and its expression is significantly increased during mouse testes development. It is mostly phosphorylated and cytosolic expressed in germ cells and promotes the exon 5-included splicing of CREB transcript (*Xiao et al., 2007*) However, in the cryptorchid testes, the dephosphorylated SRSF10 was significantly increased, and the exon 5-included CREB transcripts were increased (*Xiao et al., 2007*). Moreover, phosphorylated SRSF10 is increased during sperm maturation (*Xiao et al., 2011*). These data imply the potential importance of SRSF10 functioning as a regulator of AS in testes development and spermatogenesis, which warrants further elucidation.

In this study, we generated germ cell conditional *Srsf10* knockout mice and found that *Srsf10* was essential for spermatogenesis and male fertility. Depletion of *Srsf10* in germ cells impeded the expansion of PLZF-positive (PLZF$^+$) progenitors, followed by the failure of efficient differentiation of spermatogonia (marked by KIT expression) and meiosis initiation. Single-cell RNA-seq data confirmed that progenitors and differentiating spermatogonia were seriously lost in the *Srsf10* knockout testes at P8. The cell cycle, proliferation, and survival were impaired in the residual *Srsf10*-depleted PLZF$^+$ undifferentiated progenitors. Further analysis showed that SRSF10 directly bound thousands of genes and was involved in the AS of genes functioning in the cell cycle and germ cell development in spermatogonia. Our data reveal that SRSF10 regulates AS in spermatogonia and male fertility.

## Results

### *Srsf10* is essential for male fertility and spermatogenesis

To explore the function of *Srsf10* in male fertility and spermatogenesis, we mated *Srsf10*$^{Floxed/Floxed}$ (*Srsf10*$^{F/F}$) mice, in which the exon 3 (104 bps) was flanked by two loxP sites, with *Vasa-Cre* transgenic mice in which the *Cre* recombinase driven by a *Vasa* promoter is specifically expressed in germ cells as early as E15 (*Gallardo et al., 2007*; *Figure 1A*, *Figure 1—figure supplement 1A and B*). *Srsf10*$^{F/+}$;Vasa-Cre males and *Srsf10*$^{F/F}$ females were used for breeding, and the progeny was used for further experimental analysis (*Figure 1—figure supplement 1A and B*). Four genotypes of mice, including *Srsf10*$^{F/+}$, *Srsf10*$^{F/-}$, *Srsf10*$^{F/+}$;Vasa-Cre, and *Srsf10*$^{F/-}$;Vasa-Cre, were obtained and verified by PCR (*Figure 1—figure supplement 1B and C*). SRSF10 expression was barely detected in Mouse Vasa Homologue (MVH)-positive germ cells of *Srsf10*$^{F/-}$;Vasa-Cre (hereafter referred to as *Srsf10*-cKO) mice at postnatal day 8 (P8) (*Figure 1B*), confirming *Srsf10* was specifically depleted in germ cells at P8.

The 2-month-old *Srsf10*-cKO males looked grossly normal. Normal copulatory plugs could be observed when adult *Srsf10*-cKO males were mated with ICR wild-type females, but no pups were obtained (*Table 1*), indicating that *Srsf10*-cKO males were infertile. Compared to that of 2-month-old *Srsf10*$^{F/+}$ or *Srsf10*$^{F/+}$;Vasa-Cre mice, the testes of *Srsf10*-cKO males were much smaller (*Figure 1C*). The weight of testes and ratio of testes to body weight in *Srsf10*-cKO males were significantly reduced (*Figure 1D and E*). We then analyzed the histology of 2-month-old *Srsf10*$^{F/+}$, *Srsf10*$^{F/+}$;Vasa-Cre mice, and *Srsf10*-cKO testes using hematoxylin and eosin (H&E) staining. While all populations of spermatogonia, spermatocytes, and spermatids were observed in *Srsf10*$^{F/+}$ or *Srsf10*$^{F/+}$;Vasa-Cre mice seminiferous tubules, no spermatocytes and spermatids could be observed in the center of seminiferous tubules from *Srsf10*-cKO testes (*Figure 1F*). Moreover, no mature spermatozoa could be found in the cauda epididymis of *Srsf10*-cKO mice (*Figure 1F*). To further identify the remaining germ cells in *Srsf10*-cKO testes, we co-stained these germ cells with MVH and PLZF (the marker of all undifferentiated spermatogonia; *Buaas et al., 2004*). Immunofluorescence results showed that the MVH$^+$ and PLZF$^+$ cells were comparable in 2-month-old *Srsf10*$^{F/+}$ and *Srsf10*$^{F/+}$;Vasa-Cre mice (*Figure 1G*). As *Srsf10*$^{F/+}$ and *Srsf10*$^{F/+}$;Vasa-Cre mice were healthy and phenotypically normal, *Srsf10*$^{F/+}$;Vasa-Cre mice (hereafter, 'control') were used as control in the following experiments unless otherwise noted. Only sporadic MVH$^+$PLZF$^+$ cells can be detected around the basement of seminiferous tubules from 2-month-old *Srsf10*-cKO testes (*Figure 1G*). Statistical analysis showed that the number of PLZF$^+$ cells was significantly reduced in 2-month-old *Srsf10*-cKO testes compared to the control (*Figure 1H*), suggesting that only a few PLZF$^+$ undifferentiated spermatogonia were left in the adult *Srsf10*-cKO testes. Similarly, testes from 1-month-old *Srsf10*-cKO were also much smaller, and spermatocytes

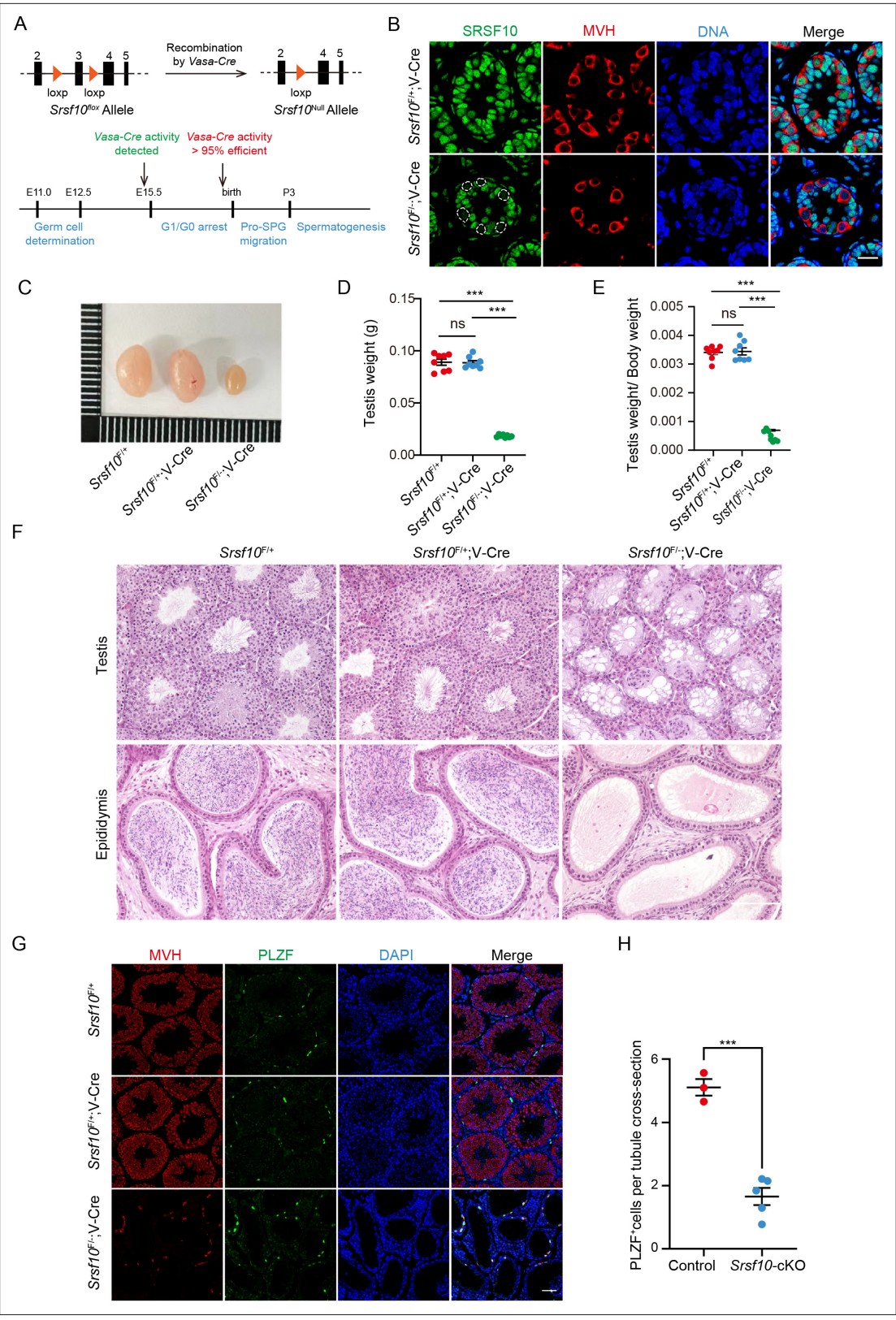

**Figure 1.** *Srsf10* is required for spermatogenesis and male fertility. (**A**) Schematic showing the deletion of *Srsf10* exon 3 and generation of *Srsf10^{F/−}*;Vasa-Cre by *Vasa-Cre*-mediated recombination in male germ cells as early as embryonic day 15 (E15). (**B**) Immunofluorescence (IF) staining for SRSF10 in the *Srsf10^{F/+}*;Vasa-Cre and *Srsf10^{F/−}*;Vasa-Cre testes of postnatal day 8 (P8) mice. White circles denote the SRSF10 null germ cells. MVH (a germ cell marker) was co-stained to indicate the location of germ cells. The DNA was stained with Hoechst 33342. Scale bar, 20 μm. (**C**) Morphological analysis

*Figure 1 continued on next page*

*Figure 1 continued*

of the testes of 2-month-old *Srsf10*$^{F/+}$, *Srsf10*$^{F/+}$;Vasa-Cre, and *Srsf10*$^{F/−}$;Vasa-Cre mice. (**D**) Testis weight of 2-month-old *Srsf10*$^{F/+}$, *Srsf10*$^{F/+}$;Vasa-Cre, and *Srsf10*$^{F/−}$;Vasa-Cre mice. Two-tailed Student's t-test was used for statistics. n=4, ***p<0.001, ns, no significance. Error bars represent s.e.m. (**E**) The ratio of testes to body weight in 2-month-old *Srsf10*$^{F/+}$, *Srsf10*$^{F/+}$;Vasa-Cre, and *Srsf10*$^{F/−}$;Vasa-Cre mice. Two-tailed Student's t-test was used for statistics. n=4, ***p<0.001, ns, no significance. Error bars represent s.e.m. (**F**) Hematoxylin and eosin (H&E) staining of testes and cauda epididymis in 2-month-old *Srsf10*$^{F/+}$, *Srsf10*$^{F/+}$;Vasa-Cre, and *Srsf10*$^{F/−}$;Vasa-Cre mice. Scale bar, 50 µm. (**G**) Co-staining for MVH (all germ cell marker) and PLZF (all undifferentiated spermatogonia marker) in adult testes in 2-month-old *Srsf10*$^{F/+}$, *Srsf10*$^{F/+}$;Vasa-Cre, and *Srsf10*$^{F/−}$;Vasa-Cre mice. DNA was stained with Hoechst 33342. Scale bar, 50 µm. (**H**) Statistics of PLZF-positive cells per tubule cross-section of testes in 2-month-old *Srsf10*$^{F/+}$;Vasa-Cre (hereafter, 'control') and *Srsf10*$^{F/−}$;Vasa-Cre (hereafter, '*Srsf10*-cKO') mice. 174 tubule cross-sections were counted in control from 3 different mice. 407 tubule cross-sections were counted in *Srsf10*-cKO from 5 different mice. Two-tailed Student's t-test was used for statistics. ***p<0.001. Error bars represent s.e.m.

The online version of this article includes the following source data and figure supplement(s) for figure 1:

**Source data 1.** The testis weight and the number of PLZF+ cells of adult male mice.

**Figure supplement 1.** Generation, breeding, and genotyping of control and *Srsf10*-cKO mice.

**Figure supplement 1—source data 1.** Raw genotyping gel of control and *Srsf10*-cKO mice using genomic PCRs.

**Figure supplement 2.** *Srsf10* is required for the first wave of spermatogenesis.

**Figure supplement 2—source data 1.** The testis weight of 1-month-old male mice.

were absent (*Figure 1—figure supplement 2*). Taken together, these data demonstrate that *Srsf10* is essential for male fertility and spermatogenesis.

## *Srsf10* depletion leads to a severe defect in meiosis initiation

As no spermatocytes were observed in the seminiferous tubules of 2-month-old and 1-month-old *Srsf10*-cKO testes, we speculated that the meiosis process may be failed in *Srsf10*-cKO mice. To figure out which stage in spermatogenesis was affected after *Srsf10* deletion, we carefully analyzed the sections from P8, P10, P12, and P15 testes using H&E staining (*Ahmed and de Rooij, 2009*). At P8, type A spermatogonia (A), intermediate spermatogonia (In), and type B spermatogonia (B) could be observed in both control and *Srsf10*-cKO testes. Afterward, some germ cells entered into meiosis and developed into leptotene spermatocytes in P10 control testes, and abundant zygotene and pachytene spermatocytes could be observed in P12 and P15 control testes (*Figure 2A*). However, rare meiotic spermatocytes were detected in the seminiferous tubules of P10, P12, and P15 *Srsf10*-cKO testes (*Figure 2A*), suggesting that meiosis initiation might be impaired in *Srsf10*-deleted mice.

To further confirm the above results, we co-stained MVH and SYCP3, which was a component of the synaptonemal complex and the marker of meiosis prophase I at P12. Spermatocytes in the vast majority of control seminiferous tubules had entered into meiotic prophase I as expected, and abundant MVH$^+$SYCP3$^+$ germ cells could be detected in the center. However, only very few seminiferous tubules contained a tiny minority (about 3–5) of MVH$^+$SYCP3$^+$ germ cells could be observed in *Srsf10*-cKO testes (*Figure 2B*). Statistically, the ratio of SYCP3$^+$ seminiferous tubule cross-sections in *Srsf10*-cKO testes was significantly lower than the control at P12 (16.41%±4.79 vs 81.45%±4.33, mean ± SEM) (*Figure 2C*). Next, we detected the expression of the marker of homologous recombination γH2AX, which can be another indicator of meiosis, in P12 testes. Immunofluorescence results showed that γH2AX$^+$ cells were nearly undetectable in the seminiferous tubules of *Srsf10*-cKO testes (*Figure 2D*). Western blot showed that γH2AX was markedly elevated in control testes from P10 to P12, consistent with the entering into meiotic prophase I of many spermatocytes around this time.

**Table 1.** The fertility of *Srsf10*-cKO males.

| Genotype | | | | | |
| --- | --- | --- | --- | --- | --- |
| Male | Female | No. of male mice | No. of plugged female mice | No. of litters | No. of pups per litter |
| Control | WT | 8 | 17 | 14 | 13.43±3.78 |
| *Srsf10*-cKO | WT | 8 | 16 | 0 | 0 |

*8-week-old wide-type ICR mice.

The online version of this article includes the following source data for table 1:

**Source data 1.** The fertility of Srsf10-cKO male mice.

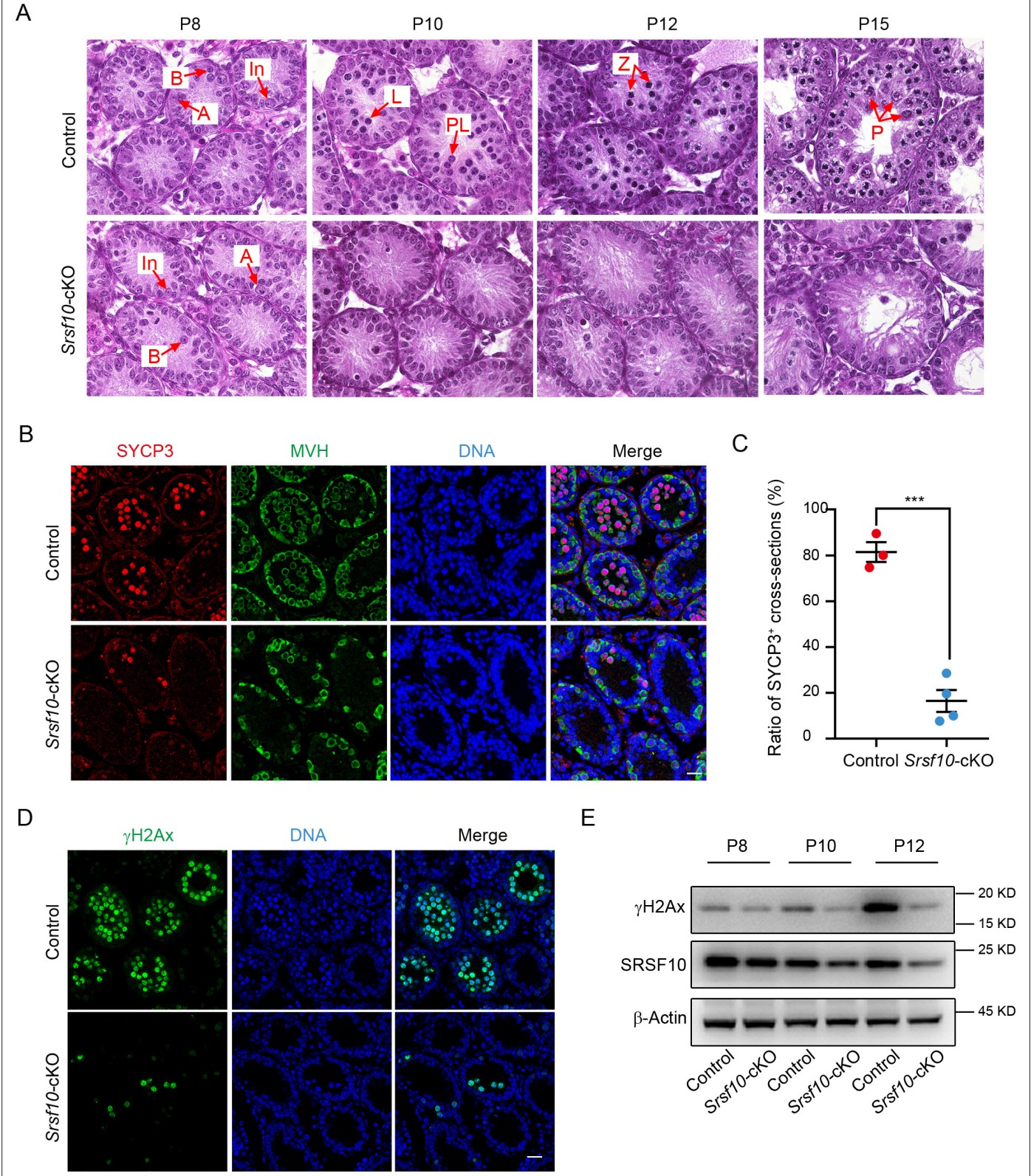

**Figure 2.** The meiosis initiation is impaired in *Srsf10*-deficient germ cells. (**A**) Hematoxylin and eosin staining of control and *Srsf10*-cKO testes at postnatal day 8 (P8), P10, P12, and P15. Spermatogenic cells were shown in cross-sections of seminiferous tubules from control and *Srsf10*-cKO testes. Spermatogenic cell types were distinguished by H&E staining according to theirchromatin morphologies (**Ahmed and de Rooij, 2009**). Red arrows indicate the representative stages of the spermatocytes. A, type A spermatogonia; In, intermediate spermatogonia; B, type B spermatogonia; L,

*Figure 2 continued on next page*

*Figure 2 continued*

leptotene spermatocytes; Z, zygotene spermatocytes; P, pachytene spermatocytes. Scale bar, 10 µm. (**B**) Immunofluorescence co-staining for SYCP3 and MVH in control and *Srsf10*-cKO testes at P12. Scale bar, 20 µm. (**C**) Statistics of the ratio of SYCP3-positive tubule cross-sections in control and *Srsf10*-cKO testes at P12. 845 tubule cross-sections were counted in control from 3 different mice. 1507 tubule cross-sections were counted in *Srsf10*-cKO from 4 different mice. Two-tailed Student's t-test was used for statistics. ***p<0.001. Error bars represent s.e.m. (**D**) Immunofluorescence staining for γH2AX in control and *Srsf10*-cKO testes at P12. Scale bar, 20 µm. (**E**) Western blot analyses of γH2AX in control and *Srsf10*-cKO testes at P8, P10, and P12. β-actin was used as the loading control. Because SRSF10 was only specifically depleted in the germ cells, the detected residual SRSF10 protein was likely from the somatic cells in the *Srsf10*-cKO testes.

The online version of this article includes the following source data for figure 2:

**Source data 1.** The ratio of SYCP3[+] tubule cross-sections in P12 testes.

**Source data 2.** RAW western blot for γH2AX, SRSF10, and β-actin.

However, the expression of γH2AX changed little in *Srsf10*-cKO testes from P8 to P12 (***Figure 2E***), indicating failure of meiosis entrance. These results indicate that *Srsf10* deletion causes a severe defect in meiosis initiation during spermatogenesis.

## *Srsf10* depletion impairs the expansion and differentiation of the progenitor spermatogonia population

Before meiosis, SSCs proliferate to self-renew or generate differentiation-committed progenitors. The progenitor spermatogonia respond to the RA signal and divide into KIT[+] differentiating spermatogonia (***Figure 3A***). Then type B differentiating spermatogonia further proliferate and differentiate into meiotic spermatocytes. Therefore, the initiation of meiosis requires several rounds of mitotic divisions of progenitors for full expansion and efficient differentiation (***Song and Wilkinson, 2014***; ***Niedenberger et al., 2015***; ***Busada et al., 2015***). Interestingly, we found that the expression of MVH and PLZF was significantly reduced in *Srsf10*-cKO testes from P8 to P12 (***Figure 3B***). Co-staining of MVH and PLZF showed that the number of PLZF[+] undifferentiated spermatogonia was apparently reduced in *Srsf10*-cKO testes at P12 (***Figure 3—figure supplement 1A and B***). These results indicate that loss of *Srsf10* affects the normal development of spermatogonia, which may further lead to the failure of meiosis initiation.

Considering the loss of PLZF[+] cells in P12 *Srsf10*-cKO testes, we speculated that the proliferation and thus the differentiation of progenitors may be affected when *Srsf10* is depleted. Therefore, we analyzed the distribution and the number of both undifferentiated and differentiating spermatogonia in control and *Srsf10*-cKO testes at P8 and P6, when only spermatogonia and Sertoli cells can be observed in the seminiferous tubules. At P8, co-staining for MVH and PLZF showed that numerous differentiating spermatogonia (MVH[+]PLZF[−]) could be detected at the basement of seminiferous tubules in control. However, MVH and PLZF were largely co-localized in *Srsf10*-cKO testes, indicative of few differentiating spermatogonia (MVH[+]PLZF[−]; ***Figure 3C***). Statistical analysis showed that differentiating spermatogonia (MVH[+]PLZF[−]) were nearly absent, and the number of undifferentiated spermatogonia (MVH[+]PLZF[+]) was significantly reduced in *Srsf10*-cKO testes compared to the control (***Figure 3C***). We further performed whole-mount staining of the testes for the differentiating spermatogonia expressing KIT. The number of KIT[+] cells was obviously reduced in *Srsf10*-cKO testes compared to the control at P8 (***Figure 3D***), further confirming the lack of differentiating spermatogonia. At P6, co-staining for MVH and PLZF showed that most of the germ cells were undifferentiated spermatogonia (MVH[+]PLZF[+]) in both control and *Srsf10*-cKO testes (***Figure 3E***). However, the number of the undifferentiated spermatogonia (MVH[+]PLZF[+]) was also significantly reduced in *Srsf10*-cKO testes compared to the control (***Figure 3E***). PLZF is a broader marker of all undifferentiated spermatogonia including As, Apr, and Aal spermatogonia (***Figure 3A***). Thus, we performed whole-mount staining for GFRα1, which is specifically expressed in As and Apr spermatogonia (***Sada et al., 2009***; ***Grasso et al., 2012***). The GFRα1[+] cells were also reduced in *Srsf10*-cKO testes compared to the control at P6 (***Figure 3F***). Altogether, these results suggest that fewer PLZF[+] undifferentiated progenitors (maybe also including GFRα1[+] SSCs) are produced in the absence of *Srsf10* as early as P6.

Because *Vasa-Cre* recombinase is expressed as early as E15.5, we examined the number of germ cells in both control and *Srsf10*-cKO testes at P3. There was no difference in germ cell numbers between control and *Srsf10*-cKO testes (***Figure 3G***). Interestingly, from P3 to P12, the number of PLZF[+] undifferentiated spermatogonia was gradually increased in both control and *Srsf10*-cKO testes.

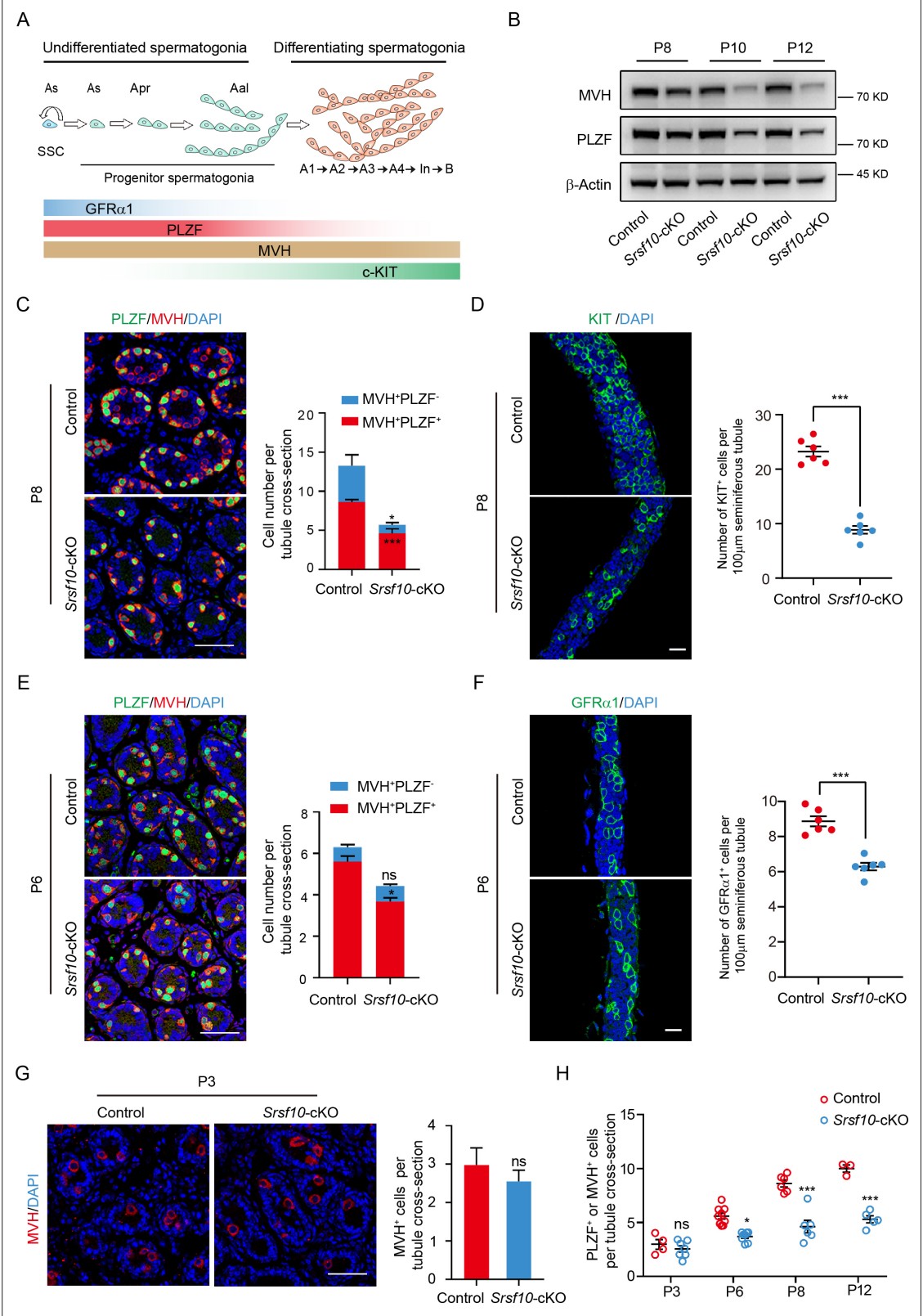

**Figure 3.** *Srsf10* depletion impairs the expansion and differentiation of progenitor spermatogonia. (**A**) Schematic showing the progression of mitosis phase of spermatogonia from spermatogonial stem cells (SSCs) to differentiating spermatogonia. The expression of representative markers at each stage is also shown. (**B**) Western blot analyses of PLZF and MVH in control and *Srsf10*-cKO testes at postnatal day 8 (P8), P10, and P12. β-actin was used as the loading control. (**C**) Left, immunofluorescence co-staining for PLZF and MVH in control and *Srsf10*-cKO testes at P8. Scale bar, 50 μm. Right,

*Figure 3 continued on next page*

*Figure 3 continued*

quantification of MVH-positive and PLZF-positive cells (MVH⁺PLZF⁺) per tubule cross-section or MVH-positive but PLZF-negative (MVH⁺PLZF⁻) cells per tubule cross-section in control and *Srsf10*-cKO testes at P8. 1286 tubule cross-sections were counted in control from 6 different mice. 1354 tubule cross-sections were counted in *Srsf10*-cKO from 6 different mice. Two-tailed Student's t-test was used for statistics. ***p<0.001. *p<0.05. Error bars represent s.e.m. (**D**) Left, whole-mount staining for KIT in control and *Srsf10*-cKO testes at P8. Scale bar, 20 μm. Right, quantification of KIT⁺ cells per 100 μm seminiferous tubule. 51 tubules were counted in control from 6 different mice. 48 tubules were counted in *Srsf10*-cKO from 6 different mice. Two-tailed Student's t-test was used for statistics. ***p<0.001. Error bars represent s.e.m. (**E**) Left, immunofluorescence co-staining for PLZF and MVH in control and *Srsf10*-cKO testes at P6. Scale bar, 50 μm. Right, quantification of MVH⁺PLZF⁻ cells or MVH⁺PLZF⁺ cells per tubule cross-section in control and *Srsf10*-cKO testes at P6. 1262 tubule cross-sections were counted in control from 3 different mice. 1069 tubule cross-sections were counted in *Srsf10*-cKO from 3 different mice. Two-tailed Student's t-test was used for statistics. *p<0.05. ns represents no significance. Error bars represent s.e.m. (**F**) Left, whole-mount staining for GFRα1 in control and *Srsf10*-cKO testes at P6. Scale bar, 20 μm. Right, quantification of GFRα1⁺ cells per 100 μm seminiferous tubule. 55 tubules were counted in control from 6 different mice. 76 tubules were counted in *Srsf10*-cKO from 6 different mice. Two-tailed Student's t-test was used for statistics. ***p<0.001. Error bars represent s.e.m. (**G**) Left, immunofluorescence staining for MVH in control and *Srsf10*-cKO testes at P3. Scale bar, 50 μm. Right, quantification of MVH-positive cells per tubule cross-section in control and *Srsf10*-cKO testes at P3. 711 tubule cross-sections were counted in control from 4 different mice. 1398 tubule cross-sections were counted in *Srsf10*-cKO from 7 different mice. Two-tailed Student's t-test was used for statistics. ns represents no significance. Error bars represent s.e.m. (**H**) Quantification of PLZF⁺ or MVH⁺ cells per tubule cross-section in control and *Srsf10*-cKO testes at P3, P6, P8, and P12. The PLZF⁺MVH⁺ cells were counted as PLZF⁺ cells at P6、 P8 and P12, but only MVH⁺ cells were counted at P3. Two-tailed Student's t-test was used for statistics. *p<0.05. ***p<0.001. ns represents no significance. Error bars represent s.e.m.

The online version of this article includes the following source data and figure supplement(s) for figure 3:

**Source data 1.** RAW western blot for MVH, PLZF, and β-actin.

**Source data 2.** Quantification of MVH⁺, MVH⁺PLZF⁺, or MVH⁺PZF⁻ cells per tubule cross-section and KIT⁺ or GFRα1⁺ cells per 100 μm seminiferous tubule.

**Figure supplement 1.** Spermatogonia development is impaired in *Srsf10*-cKO testes.

**Figure supplement 1—source data 1.** Quantification of PLZF⁺ cells per tubule cross-section in P12 control and *Srsf10*-cKO testes.

**Figure supplement 1—source data 2.** RAW western blot for PLZF and β-actin.

However, the kinetics in *Srsf10*-cKO testes was much slower, leading to a comparable number of PLZF⁺ cells per tubule cross-section in P12 *Srsf10*-cKO to that in P6 control testes (*Figure 3H*). Western blot confirmed that the expression of PLZF in P3 testes was lower, and no significant difference was observed between control and *Srsf10*-cKO testes. With the resumption of mitosis, the expression of PLZF was apparently increased from P3 to P8 in control testes but was only slightly increased in *Srsf10*-cKO testes (*Figure 3—figure supplement 1C*). All these data suggest that *Srsf10* may be essential for the expansion of the progenitor spermatogonia population, disturbance of which may further lead to differentiation defect.

## *Srsf10* depletion disturbs the expression of genes in progenitor spermatogonia

To systematically investigate the molecular effects of *Srsf10* loss in germ cells, we compared the transcriptomes of control and *Srsf10*-cKO testes at P3, P6, and P8. RNA-seq analyses identified only 10 upregulated and 9 downregulated genes in *Srsf10*-cKO testes compared with the control at P3 (fragments per kilobase of exon per million reads mapped [FPKM] ≥5, fold change ≥2, and p<0.01; *Figure 4A*), suggesting that the transcriptome is largely normal at P3 in *Srsf10*-depleted testes. This is consistent with the immunostaining result showing that the number of germ cells is comparable between control and *Srsf10*-cKO testes at P3. However, 138 genes were differentially expressed between the control and *Srsf10*-cKO testes at P6, and nearly all genes (135/138) were downregulated in *Srsf10*-cKO testes (*Figure 4A*). 396 genes were differentially expressed in *Srsf10*-cKO testes at P8, including 243 upregulated and 153 downregulated genes (*Figure 4A*). Surprisingly, only 57 genes were commonly downregulated in the *Srsf10*-cKO testes at both P6 and P8 (*Figure 4B*). GO analysis of the 57 genes showed that these genes are involved in the meiotic cell cycle, spermatogenesis, cell differentiation, and germ cell development (*Figure 4B*). We then systematically analyzed the expression pattern of all downregulated genes in control and *Srsf10*-cKO testes from P3 to P8. The vast majority of downregulated genes were gradually increased from P3 to P8 (*Figure 4—figure supplement 1A*). Interestingly, many genes that were only significantly downregulated at P6 (e.g. *Pou5f1, Sohlh1, Egr4, Nanos3, Foxc2*, and *Sox3*) were mainly involved in the expansion and early differentiation of progenitor spermatogonia (*Figure 4B*). The expression level of these genes was very

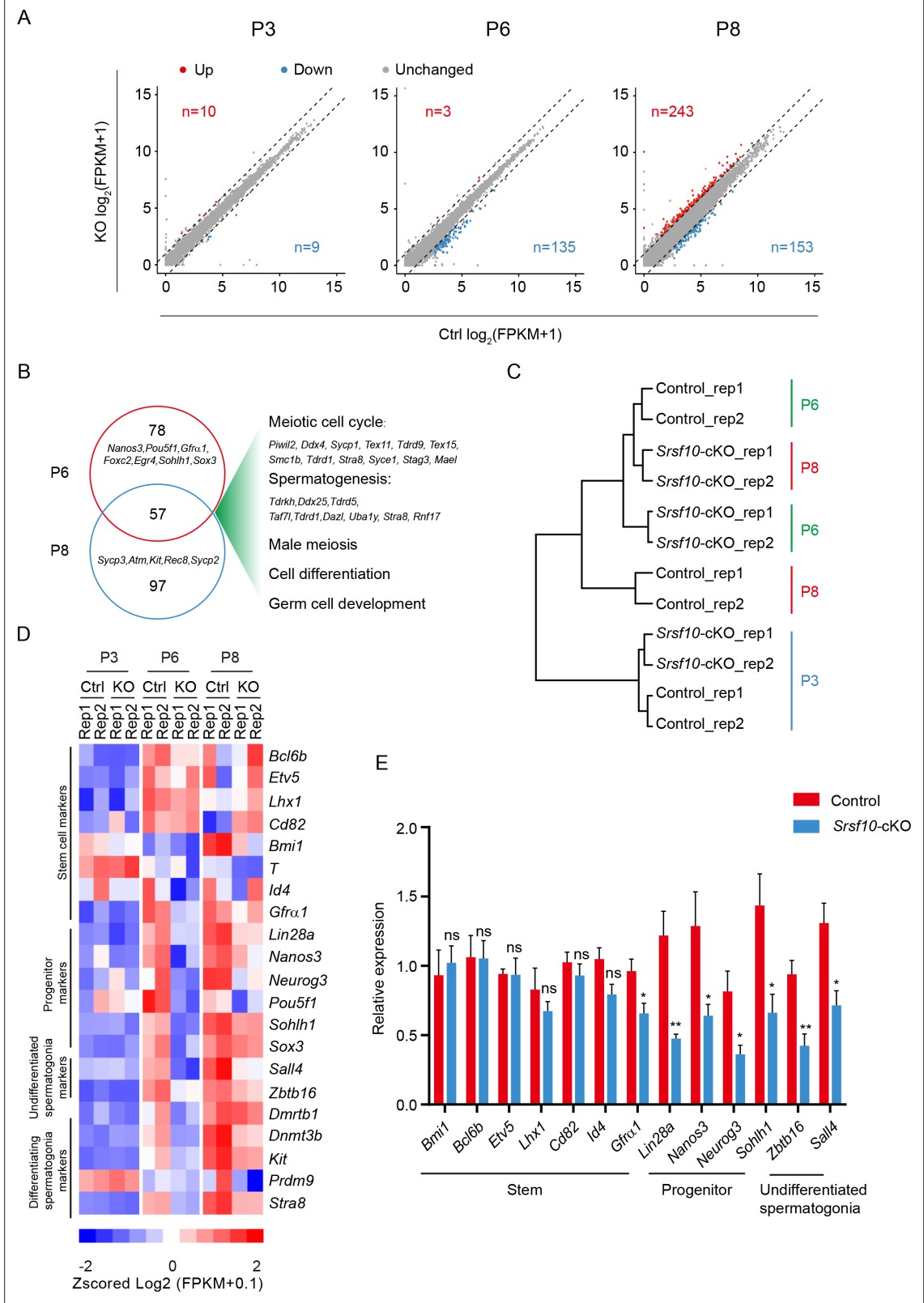

**Figure 4.** *Srsf10* deficiency alters expression patterns of genes involved in progenitor spermatogonia. (**A**) Scatter plots showing the expression of genes in control and *Srsf10*-cKO testes at postnatal day 3 (P3), P6, and P8. Blue dots represent significantly downregulated genes, while red dots show significantly upregulated genes ( fragments per kilobase of exon per million reads mapped [FPKM] ≥5, fold change ≥2, and p<0.01). Gray dots represent unchanged genes. (**B**) Venn diagram depicting the overlap of downregulated genes between P6 and P8. Gene ontology (GO) terms of the

*Figure 4 continued on next page*

*Figure 4 continued*

57 shared downregulated genes in P6 and P8 *Srsf10*-cKO testes, and genes involved in specific GO terms were shown on the right. (**C**) Hierarchical clustering of two replicates of control and *Srsf10*-cKO testes at P3, P6, and P8 based on the expression of spermatogonia-specific genes (***Green et al., 2018***). Note the closer relationship between P8 *Srsf10*-cKO and P6 control testes. (**D**) Heatmap showing the mRNA abundance of genes functioning in spermatogonial stem cells (SSCs; *Bcl6b*, *Etv5*, *Lhx1*, *Cd82*, *Bmi1*, *T*, *Id4*, and *Gfra1*), progenitors (*Lin28a*, *Nanos3*, *Neurog3*, *Pou5f1*, *Sohlh1*, and *Sox3*), undifferentiated spermatogonia (*Sall4* and *Zbtb16*), and differentiating spermatogonia (*Dmrtb1*, *Dnmt3b*, *Kit*, *Stra8*, and *Prdm9*). (**E**) Quantitative RT-PCR validation of the expression of genes involved in SSC, progenitors, undifferentiated spermatogonia, and differentiating spermatogonia in control and *Srsf10*-cKO testes at P6. β-actin was used as the internal control. Two-tailed Student's t-test was used for statistics. *p<0.05, **p<0.01, ***p<0.001; ns, no significance. Error bars represent s.e.m.

The online version of this article includes the following source data and figure supplement(s) for figure 4:

**Source data 1.** List of the differential expression genes in control and *Srsf10*-cKO testes at P3, P6, and P8, respectively.

**Source data 2.** Validation of the expression of marker genes in control and *Srsf10*-cKO testes using qRT-PCR at P6.

**Figure supplement 1.** Systematical analysis of the downregulated genes from postnatal day 3 (P3) to P8.

low in P6 *Srsf10*-cKO testes and somewhat upregulated afterward in P8 *Srsf10*-cKO testes (***Figure 4—figure supplement 1A***). Many genes that were only significantly downregulated at P8 (e.g. *Kit*, *Sycp3*, *Sycp2*, *Atm,* and *Rec8*) were involved in spermatogonia differentiation and meiosis and were barely upregulated in *Srsf10*-cKO testes from P6 to P8 (***Figure 4B*** and ***Figure 4—figure supplement 1A***), echoing nearly no differentiating spermatogonia and meiotic spermatocytes in P8 *Srsf10*-cKO testes. Unsupervised cluster analysis of spermatogonia-specific genes (SPGs; ***Green et al., 2018***) showed that the gene expression pattern of *Srsf10*-cKO testes at P8 was much more similar to that of control testes at P6 (***Figure 4C***), consistent with the compromised and slower expansion and differentiation of progenitor spermatogonia in the *Srsf10*-cKO testes than control.

We then analyzed several marker genes for stem cell maintenance, expansion, and differentiation of the spermatogonia population. Most genes involved in SSCs maintenance (*Etv5*, *Bmi1*, *Id4*, *Lhx1*, *Cd82,* and *T*) were hardly affected in the *Srsf10*-cKO testes from P3 to P8. However, genes associated with progenitor spermatogonia (*Lin28a*, *Nanos3*, *Sox3*, *Neurog3*, *Pou5f1*, and *Sohlh1*), undifferentiated spermatogonia (*Zbtb16* (*Plzf*) and *Sall4*), and differentiating spermatogonia (*Dmrtb1*, *Dnmt3b*, *Stra8*, *Kit*, and *Prdm9*) were downregulated in *Srsf10*-cKO testes at P6 and P8 compared to control (***Figure 4D*** and ***Figure 4—figure supplement 1B***). The qRT-PCR results also confirmed that the expression of genes involved in the expansion and differentiation of progenitor cells was globally reduced in P6 *Srsf10*-cKO testes (***Figure 4E***). Taken together, these data suggest that *Srsf10* deficiency disturbs genes involved in progenitor spermatogonia and thus their expansion and differentiation.

## Single-cell RNA-seq reveals that the progenitor and differentiating spermatogonia are largely lost in *Srsf10*-cKO testes

To address the heterogeneity of spermatogonia after *Srsf10* depletion, we sought to enrich the undifferentiated and differentiating spermatogonia using their cell surface markers, THY1 (undifferentiated spermatogonia) and KIT (differentiating spermatogonia). We enriched THY1⁺ KIT⁻, THY1⁻KIT⁺, and THY1⁺ KIT⁺ spermatogonia from control (one replicate isolated from three different mice) and *Srsf10*-cKO testes (two replicates isolated from six different mice) using magnetic-activated cell sorting (MACS) with anti-THY1 and anti-KIT antibodies at P8 (***Oatley et al., 2006***). The isolated single cells were subjected to single-cell RNA-seq analysis using the 10× Genomics platform (***Figure 5A***). After filtering out low-quality cells, 1157 control and 766 *Srsf10*-cKO spermatogonial cells were used for further analysis. Clustering analysis identified five distinct spermatogonial subtypes/states as visualized using uniform manifold approximation and projection (UMAP; ***Becht et al., 2018***; ***Figure 5B***). Based on known cell type markers, we named these cell clusters USPG1 (undifferentiated spermatogonia 1), USPG2, DSPG1 (differentiating spermatogonia 1), DSPG2, and DSPG3 (***Figure 5B***). USPG1 cells corresponded to SSCs, as they highly express SSC marker genes, including *Eomes*, *Id4*, *Lhx1*, *Etv5*, *Gfra1*, and *Smoc2* (***Figure 5C*** and ***Figure 5—figure supplement 1***). USPG2 cells were highly enriched for *Nanos3*, *Pou5f1,* and *Zbtb16* (*Plzf*), with increased levels of *Sox3* and *Sohlh1*, but lower levels of SSC-associated markers and differentiation markers (***Figure 5C*** and ***Figure 5—figure supplement 1***), corresponding to early progenitor spermatogonia. By contrast, DSPG1 cells expressed high levels of early differentiation marker genes such as *Stra8*, *Sox3*, *Sohlh1*, *Sohlh2*, and *Dmrt1* and low levels

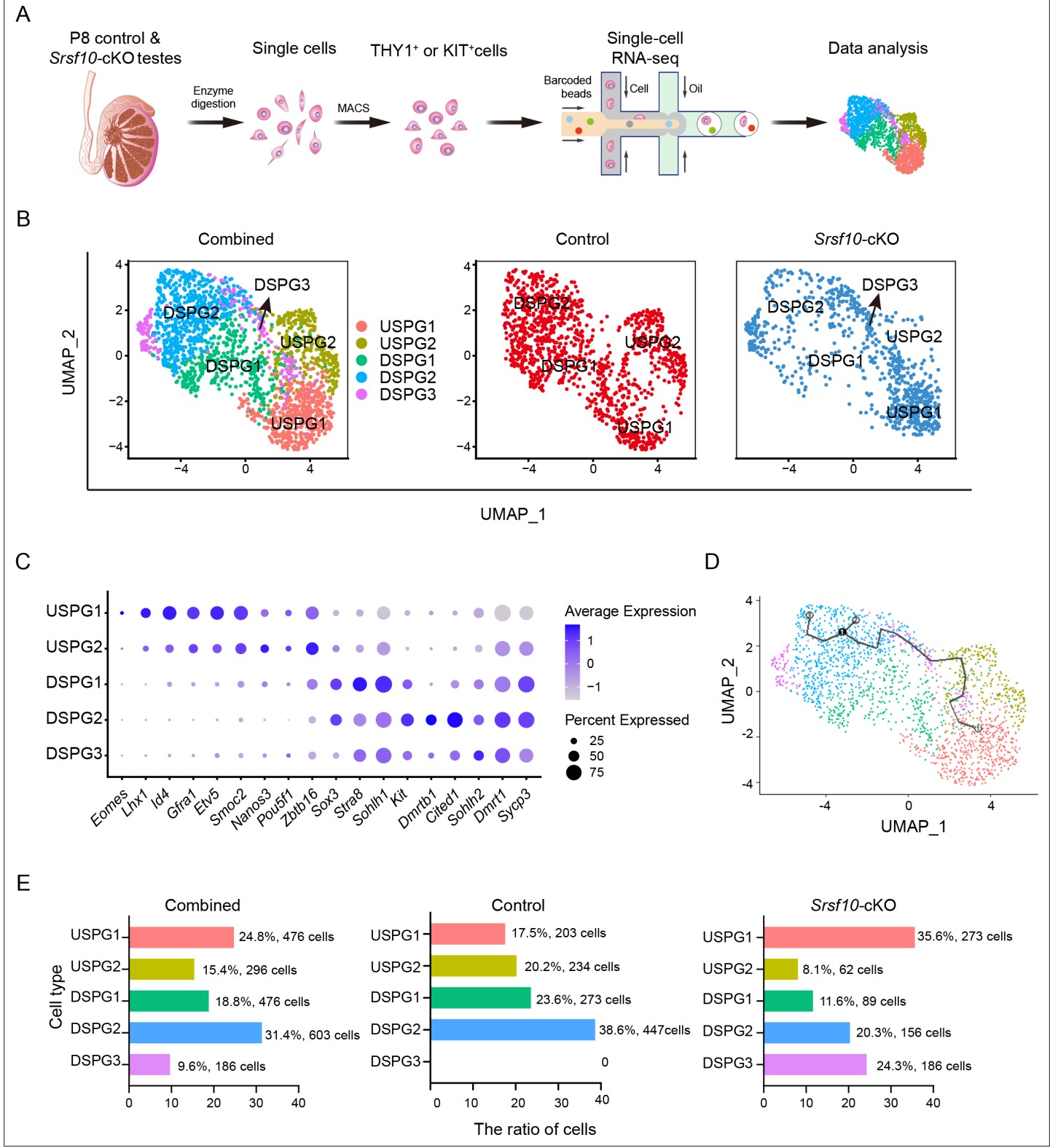

**Figure 5.** Single-cell RNA sequencing (scRNA-seq) defines the transcriptome-wide signatures of spermatogonia development in *Srsf10*-cKO testes at postnatal day 8 (P8). (**A**) Schematic illustration of the workflow for scRNA-seq analysis. (**B**) Uniform manifold approximation and projection (UMAP) clustering analysis of combined (left), control (middle), and *Srsf10*-cKO (right) spermatogonia. Five subtypes, including USPG1 (undifferentiated spermatogonia 1), USPG2, DSPG1 (differentiating spermatogonia 1), DSPG2, and DSPG3 are identified and color-coded in the left panel. (**C**) Dot plot

*Figure 5 continued on next page*

*Figure 5 continued*

showing the expression of selected marker genes across the five spermatogonia subtypes. (**D**) Pseudotime trajectory analysis of the indicated cell clusters. (**E**) Quantification of the number and percentage of spermatogonia in each subtype in combined, control, and *Srsf10*-cKO groups.

The online version of this article includes the following figure supplement(s) for figure 5:

**Figure supplement 1.** The expression pattern of spermatogonial marker genes in the subtypes of control and *Srsf10*-cKO samples.

of *Pou5f1, Nanos3,* and *Zbtb16* (*Figure 5C* and *Figure 5—figure supplement 1*), indicating that this cell population is likely late progenitor/early differentiating spermatogonia. DSPG2 cells displayed lower *Stra8* expression but mainly expressed differentiation marker genes including *Kit, Dmrtb1, Cited1,* and *Dmrt1* (*Figure 5C* and *Figure 5—figure supplement 1*), implicating them as differentiated spermatogonia. The DSPG3 cluster was only observed from the *Srsf10*-cKO group. These cells express some differentiating markers such as *Stra8, Dmrt1, Sohlh2,* and *Kit,* but the expression level of *Stra8* and *Kit* was lower than DSPG1 and DSPG2, respectively (*Figure 5C*), suggesting that DSPG3 cells might have differentiated but with abnormal development. The pseudotime analysis provided a trajectory indicating the development of germline cells from USPG1 to DSPG2 (*Figure 5D*).

We then asked whether and how the subtypes of spermatogonia changed after *Srsf10* depletion. We found that USPG1 (17.5%, 203 cells), USPG2 (20.2%, 234 cells), DSPG1 (23.6%, 273 cells), and DSPG2 cells (38.6%, 447 cells) were largely evenly distributed in the control group (*Figure 5E*). By contrast, the proportion in the *Srsf10*-cKO group was severely slanted, which was manifested by the dramatic decrease of USPG2 (8.1%, 62 cells), DSPG1 (11.6%, 89 cells), and DSPG2 (20.3%, 156 cells) subtypes, and the dominance of the USPG1 subtype (35.6%, 273 cells). On the other hand, 24.3% (186 cells) of the cells were sorted into the DSPG3 subtype, which was scattered in the other 4 spermatogonial subtypes (*Figure 5E*). These data suggest that *Srsf10* deficiency severely impairs the development of spermatogonia. Taken together, single-cell transcriptional analysis of spermatogonia shows that the progenitor and differentiating spermatogonia are largely lost in *Srsf10*-cKO testes at P8, further confirming the requirement of *Srsf10* for the expansion of progenitor spermatogonia.

## The cell cycle and proliferation of spermatogonia are impaired in *Srsf10*-cKO testes

We then analyzed the differentially expressed genes in the aforementioned four subtypes of cells between the control and *Srsf10*-cKO samples. In USPG1, 184 and 175 genes were downregulated and upregulated, respectively (|log2FC|>0.25 and p<0.01; *Figure 6A*). The downregulated genes were mainly enriched for cell cycle, chromosome segregation, mitotic spindle organization, spermatogenesis, and cell proliferation. The upregulated genes were enriched for G1/S transition of mitotic cell cycle, regulation of apoptotic process, and negative regulation of cell proliferation (*Figure 6B*). Consistently, *Top2a, Mki67, Cdc20, Ccnb1, Ccna2, Cenpe, Rad50, Kif11, Nusap1,* and *Prc1,* which are important for cell cycle and cell division, were significantly reduced in the USPG1 cells in the *Srsf10*-cKO sample (*Figure 6B and C* and *Figure 6—figure supplement 1*). *Dnaja1, Aspm, Ccnb1, Btg1, Racgap1, Sycp1, Sycp3,* and *Mns1,* which are required in spermatogenesis, were also significantly reduced in the USPG1 cells in the absence of *Srsf10* (*Figure 6B and C* and *Figure 6—figure supplement 1*). In the USPG2, DSPG1, and DSPG2, the downregulated genes were also enriched for cell cycle, cell division, and cell proliferation (*Figure 6—figure supplement 2*). These data indicate that depletion of *Srsf10* in germ cells likely disrupts the normal cell cycle and proliferation of spermatogonia.

To test this idea, we first investigated the proliferation of spermatogonia in *Srsf10*-cKO testes by co-staining for PLZF and Ki-67 (labeling mitotic cells that are not in the G0 phase) to analyze the mitotic status of spermatogonia at P6. The immunofluorescence result showed nearly all PLZF⁺ cells were positive for Ki-67 in both control and *Srsf10*-cKO testes (*Figure 6D*). The number of Ki-67⁺PLZF⁺ cells was not significantly different between control and *Srsf10*-cKO testes (*Figure 6E*), suggesting that all the PLZF⁺ cells were in the mitotic status. We further tested the EdU incorporation of PLZF⁺ spermatogonia in the control and *Srsf10*-cKO testes at P6. EdU is a thymidine analog and can incorporate into the DNA molecule during S-phase. Control and *Srsf10*-cKO mice were intraperitoneally injected with EdU (5 mg/kg), and testes were collected 4 hr later for fixation and paraffin section. Co-staining for EdU and PLZF showed that the EdU incorporation was significantly reduced in *Srsf10*-cKO PLZF⁺

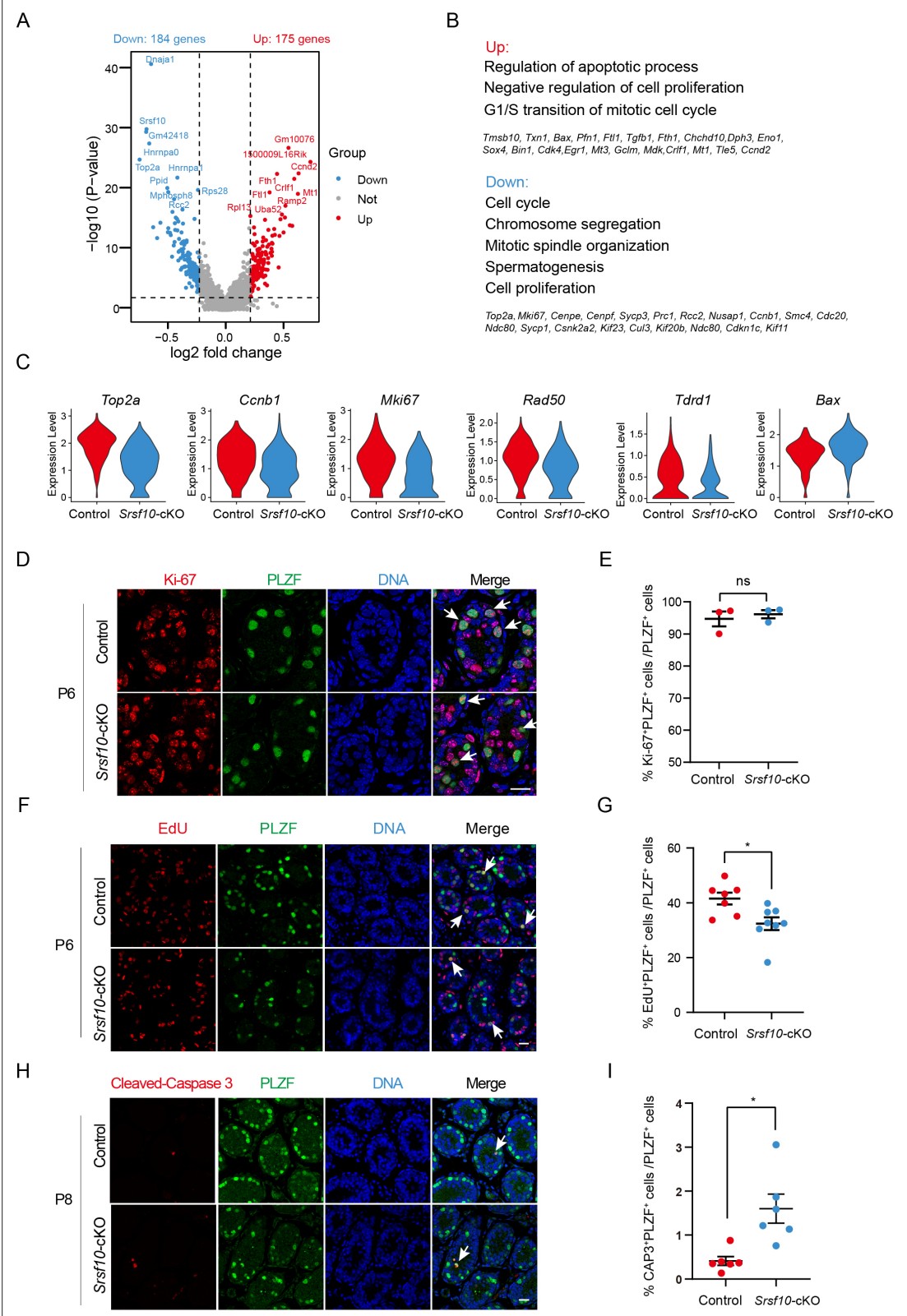

**Figure 6.** *Srsf10* depletion impairs the cell cycle and proliferation of undifferentiated spermatogonia. (**A**) Volcano plot showing the significantly differentially expressed transcripts in the USPG1 subtype in *Srsf10*-cKO compared with the control samples. Blue dots represent significantly downregulated transcripts, while red dots show significantly upregulated transcripts (|log2FC|>0.25 and p<0.01). Gray dots represent unchanged transcripts. (**B**) Gene ontology of upregulated and downregulated genes in *Srsf10*-cKO USPG1 subtype and representative genes in up- and

*Figure 6 continued on next page*

*Figure 6 continued*

downregulated groups are shown. (**C**) Violin plots showing the expression of functional genes involved in cell cycle and spermatogenesis in the USPG 1 subtype in control and *Srsf10*-cKO groups. (**D**) Immunofluorescence co-staining for the mitosis marker Ki-67 and PLZF in control and *Srsf10*-cKO testes at postnatal day 6 (P6). The DNA was stained with Hoechst 33342. Double-positive cells (Ki-67⁺PLZF⁺) are indicated by the white arrowhead. Scale bar, 50 µm. (**E**) Quantification of the ratio of Ki-67⁺PLZF⁺ cells in PLZF⁺ cells in control and *Srsf10*-cKO testes at P6. 841 PLZF⁺ cells were counted in control from 3 different mice. 624 PLZF⁺ cells were counted in *Srsf10*-cKO from 3 different mice. Two-tailed Student's t-test was used for statistics. ns, no significance. Error bars represent s.e.m. (**F**) Immunofluorescence co-staining for the EdU and PLZF in control and *Srsf10*-cKO testes at P6. Control and *Srsf10*-cKO mice were treated with EdU for 4 hr. The DNA was stained with Hoechst 33342. White arrowheads indicate the representative EdU⁺PLZF⁺ cells. Scale bar, 20 µm. (**G**) Quantification of the ratio of EdU⁺PLZF⁺ cells in PLZF⁺ cells of control and *Srsf10*-cKO testes at P6. 4356 PLZF⁺ cells were counted in control from 7 different mice. 2739 PLZF⁺ cells were counted in *Srsf10*-cKO from 8 different mice. Two-tailed Student's t-test was used for statistics. *p<0.05, Error bars represent s.e.m. (**H**) Immunofluorescence co-staining for the apoptosis marker cleaved caspase 3 (CAP3) and PLZF in control and *Srsf10*-cKO testes at P8. The DNA was stained with Hoechst 33342. Scale bar, 50 µm. (**I**) Quantification of the ratio of CAP3⁺PLZF⁺ cells in PLZF⁺ cells of control and *Srsf10*-cKO testes at P8. 8298 PLZF⁺ cells were counted in control from 6 different mice. 5031 PLZF⁺ cells were counted in *Srsf10*-cKO from 6 different mice. Two-tailed Student's t-test was used for statistics. *p<0.05, Error bars represent s.e.m.

The online version of this article includes the following source data and figure supplement(s) for figure 6:

**Source data 1.** List of the differential expression genes in USPG1 and GO analysis of these differential expression genes.

**Source data 2.** Quantification of the ratio of KI67⁺PLZF⁺ cells, EdU⁺PLZF⁺ cells, and CAP3⁺PLZF⁺ cells in PLZF⁺ cells in control and *Srsf10*-cKO testes.

**Figure supplement 1.** The expression pattern of key genes of the USPG1 subtype in control and *Srsf10*-cKO groups.

**Figure supplement 2.** Identification of the differential expression genes (DEGs) and gene ontology (GO) analysis of DEGs in USPG2, DSPG1, and DSPG2 subtypes between control and *Srsf10*-cKO samples.

**Figure supplement 2—source data 1.** Lists of the differential expression genes in USPG2, DSPG1, and DSPG2 and GO analysis of these differential expression genes.

**Figure supplement 3.** The proliferation of GFRα1-positive spermatogonial stem cells in control and *Srsf10*-cKO testes at P3.

**Figure supplement 3—source data 1.** Quantification of the ratio of EdU⁺GFRα1⁺ cells and in GFRα1⁺ cells in control and *Srsf10*-cKO testes at P3.

spermatogonia at P6 (***Figure 6F and G***). These results suggest that the proliferation of PLZF⁺ spermatogonia in *Srsf10*-cKO mice was significantly impaired. We also detected the EdU incorporation in spermatogonia expressing GFRα1 in the control and *Srsf10*-cKO testes at P3. Co-staining for EdU and GFRα1 showed that EdU incorporation of GFRα1⁺ cells was comparable in control and *Srsf10*-cKO P3 testes (***Figure 6—figure supplement 3***), suggesting that the proliferation of GFRα1⁺ SSCs was probably not much affected at P3.

As upregulation of apoptosis-related genes in the USPG1 and USPG2 cells, we also analyzed the apoptosis of PLZF⁺ spermatogonia. Double staining for cleaved caspase 3 (CAP3) and PLZF showed that the number of CAP3⁺PLZF⁺ spermatogonia was significantly higher in *Srsf10*-cKO testes compared to the control (***Figure 6H and I***), implying that the survival of spermatogonia was also impaired when *Srsf10* is depleted. In conclusion, *Srsf10* depletion might lead to abnormal cell cycle and mitotic division, and further affect the proliferation and survival of spermatogonia.

## Genome-wide analysis of SRSF10-binding genes in spermatogonia

We then sought to probe into how *Srsf10* depletion affects spermatogenesis at the molecular level. To this end, we collected the THY1⁺ spermatogonia using MACS from control and *Srsf10*-cKO testes at P6 to minimize the potential side effects (***Oatley et al., 2006***; ***Figure 7A***). Flow cytometry and immunostaining analyses showed that the proportion of undifferentiated spermatogonia (THY1⁺ or PLZF⁺ cells) was significantly increased in the sorted group than that in the unsorted group (***Figure 7—figure supplement 1A and B***). We confirmed the dramatic decrease of *Srsf10* in the enriched spermatogonia of *Srsf10*-cKO testes at both RNA and protein levels (***Figure 7—figure supplement 1C and D***). Next, we performed the linear amplification of complementary DNA ends and sequencing (LACE-seq) (***Figure 7A***; ***Su et al., 2021***), a low-input method for global profiling of RBP target sites, to profile the SRSF10 binding events in the enriched spermatogonia. The LACE-seq data were highly reproducible between the two replications for SRSF10 (Pearson = 0.73) (***Figure 7B***, ***Figure 7—figure supplement 2A***), which were pooled together for the following analyses. Most of the SRSF10 binding sites were in gene body regions, with 60.49% mapped to the exonic sequences (***Figure 7C***). SRSF10-binding peaks in mouse spermatogonia were significantly enriched for GA-rich consensus motif (GAAGAG) (***Figure 7D***), which was consistent with the motif analysis of RNA-seq data in chicken DT40 cells ***Zhou et al., 2014b***. Further analysis showed that SRSF10 tended to bind exons and was enriched between 0

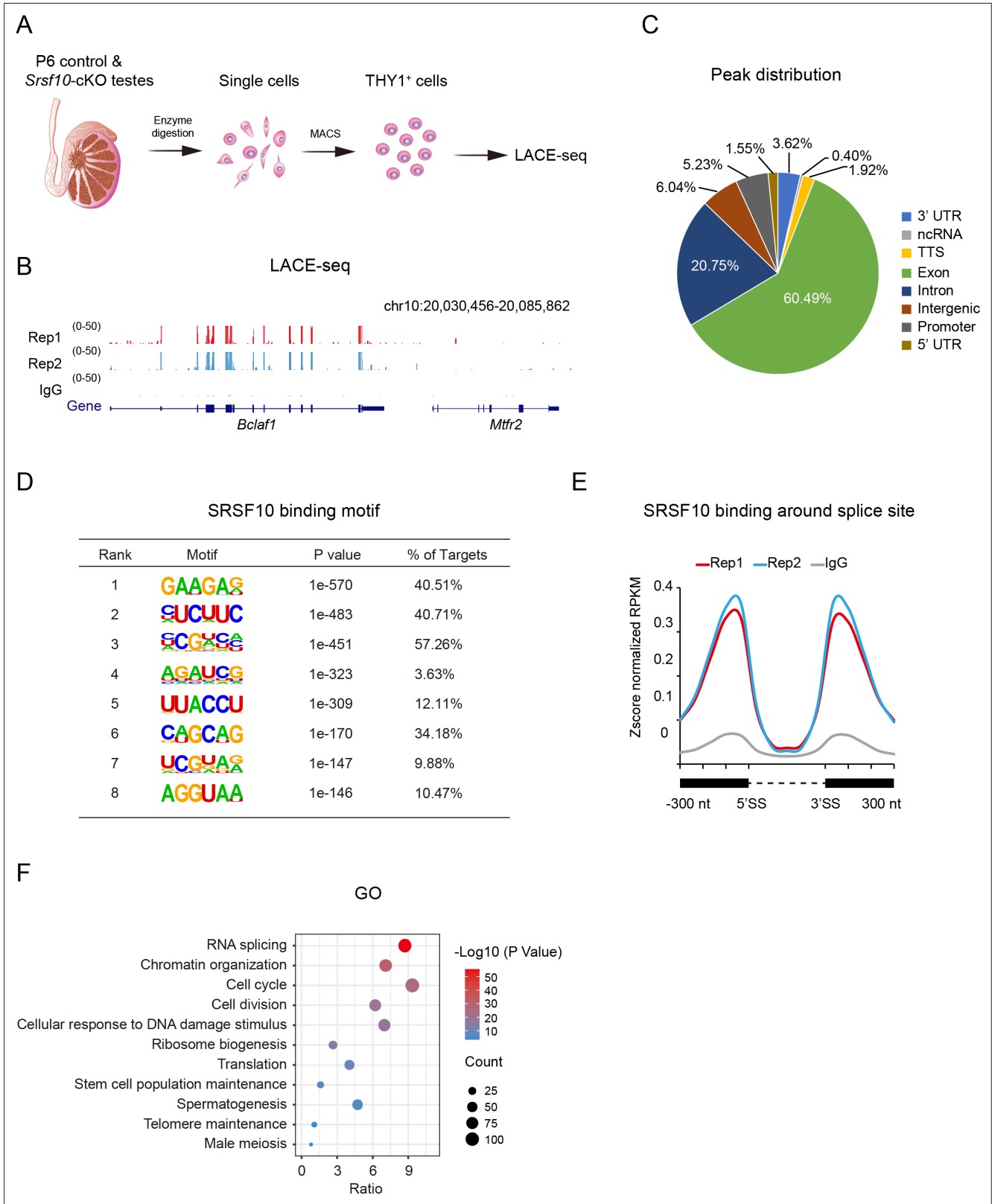

**Figure 7.** Genome-wide binding of SRSF10 in mouse spermatogonia. (**A**) Schematic showing the workflow for the enrichment of THY1+ spermatogonia from wide-type postnatal day 6 (P6) testes for linear amplification of complementary DNA ends and sequencing (LACE-seq). (**B**) The UCSC genome browser showing the reproducibility of SRSF10 LACE-seq data. IgG was used as a negative control. Note the binding of SRSF10 to *Bclaf1* but not the adjacent *Mtfr2*. (**C**) Pie chart showing the distribution of SRSF10 peaks in the genome. UTR, untranslated region; ncRNA, non-coding RNA; TTS,

*Figure 7 continued on next page*

Figure 7 continued

transcriptional termination site. (**D**) Enriched hexamer motifs bound by SRSF10. The top eight enriched motifs are shown. (**E**) Line chart showing the distribution of SRSF10 binding sites in the vicinity of the 5' exon-intron and the 3' intron-exon boundaries (300 nt upstream of 5'SS and 300 nt downstream of 3'SS). The black boxes denote exons, and the dotted line denotes intron. SS, splice site. (**F**) Gene ontology (GO) analysis of SFSR10 bound genes. The overrepresented terms, gene count, and p-value are shown.

The online version of this article includes the following source data and figure supplement(s) for figure 7:

**Source data 1.** List of SRSF10 binding targets of LACE-seq data and GO analysis of the SRSF10 binding targets of LACE-seq data.

**Figure supplement 1.** Sorting strategy for THY1+ spermatogonia in control and *Srsf10*-cKO testes at postnatal day 6 (P6).

**Figure supplement 1—source data 1.** Relative mRNA expression of *Srsf10* in enriched cells.

**Figure supplement 2.** Reproducibility of linear amplification of complementary DNA ends and sequencing (LACE-seq) data.

and 100 nt of the 5' and 3' exonic sequences flanking the constitutive splice sites (**Figure 7E**), presumably reflecting its roles in splicing. With a cutoff of FPKM ≥10, fold change ≥2 in both replicates, 1207 SRSF1-binding genes in THY1+ spermatogonia were identified (**Figure 7—figure supplement 2B**). These genes were significantly involved in chromatin organization, cell cycle, cellular response to DNA damage stimulus , stem cell population maintenance, spermatogenesis, and male meiosis (**Figure 7F**). These data indicate that SRSF10 functions as a crucial regulator by directly binding its targets in the development of spermatogonia.

## *Srsf10* binds and regulates AS of functional genes in spermatogonia

To explore how SRSF10 is involved in gene regulation in spermatogonia development, we performed next-generation sequencing of the enriched THY1+ spermatogonia of P6 control and *Srsf10*-cKO testes (**Figure 8A**) and compared the transcriptome between them. RNA-seq analyses identified only 11 downregulated genes and 60 upregulated genes in *Srsf10*-cKO spermatogonia compared with the control using a commonly used cutoff (FPKM ≥5, |log2FC|≥1, and p<0.05). But if a lower cutoff (FPKM ≥2, |log2FC|≥0.5, and p<0.05) was used, 206 downregulated genes can be identified that were significantly enriched for spermatogenesis, meiotic cell cycle, cell differentiation, and male germline stem cell asymmetric division (**Figure 8—figure supplement 1A**). Among the 206 down-regulated genes were the markers of progenitor spermatogonia, such as *Nanos3, Sox3, Lin28a,* and *Neurog3* (**Figure 8—figure supplement 1A**), presumably due to the absence of the progenitor spermatogonia, as observed in the scRNA-seq data. We compared the differentially expressed genes with 1207 targets in LACE-seq data and found that only 24 genes were overlapped (**Figure 8—figure supplement 1B**), suggesting that changes in gene expression might be regulated indirectly by SRSF10.

*Srsf10* is an SR protein and is involved in AS. We asked whether depletion of *Srsf10* would affect the splicing and AS in spermatogonia. THY1+ spermatogonia at P6 were collected and subjected to PacBio isoform sequencing (Iso-seq) (**Figure 8A**), which can generate and capture the full-length cDNAs and present more accurate information about isoforms, alternatively spliced exons, and fused transcripts. We then analyzed the Iso-seq data for identification of the differentially spliced events (DSEs) between control and *Srsf10*-cKO spermatogonia using SUPPA2 software (**Trincado et al., 2018**). 522 DSEs in 417 genes were identified (p<0.05 and |ΔPSI|>0.1), the majority of which were exon skipping (191) (**Figure 8—figure supplement 2A**). Among the 522 DSEs, 266 AS events had negative ΔPSI values, and 256 had positive ΔPSI values, indicating that *Srsf10* has no bias in the regulation of exon inclusion or exclusion (**Figure 8—figure supplement 2B**). Additionally, the RNA-seq data were analyzed for AS using Comprehensive AS Hunting (CASH) software, which is an AS event detecting method with a high rate of validation (**Wu et al., 2018**). 332 DSEs involved in 317 genes were identified (p<0.05), the majority of which were exon skipping (229) (**Figure 8—figure supplement 2C**). Combining the Iso-seq and RNA-seq data, we identified 675 abnormally spliced genes in the *Srsf10*-cKO spermatogonia (**Figure 8B**). GO analysis of the 675 genes showed that these genes were involved in cellular response to DNA damage stimulus, cell cycle, translation, mitotic G2/M transition checkpoint, DNA replication, and spermatogenesis (**Figure 8—figure supplement 2D**). These data indicate that SRSF10 functions as an important splicing regulator in mouse spermatogenesis. However, the functional consequences of these DSEs for genes themselves and specific cellular or developmental events warrant further investigation experimentally.

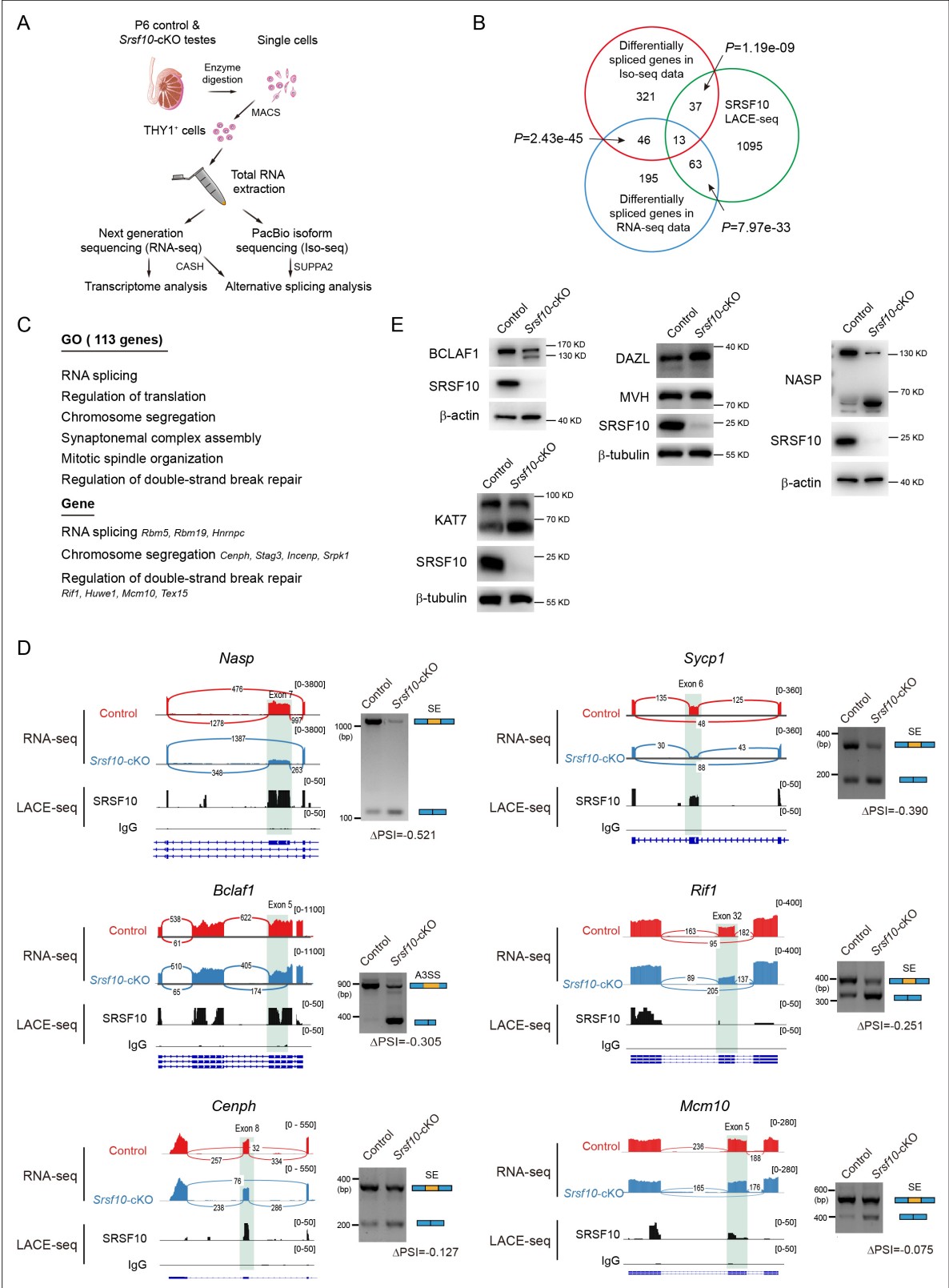

**Figure 8.** *Srsf10* is required for normal splicing of functional genes in spermatogonia. (**A**) Schematic illustration of the workflow for the enrichment of THY1+ spermatogonia for transcriptome analysis and alternative splicing (AS) analysis using isoform sequencing (Iso-seq). (**B**) Venn diagram showing the overlapped gene between differentially spliced genes (675 genes including 417 genes in Iso-seq and 317 genes in RNA-seq) and linear amplification of complementary DNA ends and sequencing (LACE-seq; 1207 genes) data. (**C**) Gene ontology (GO) terms for 113 genes directly bound by SRSF10 with

*Figure 8 continued on next page*

*Figure 8 continued*

significantly affected AS events and representative genes are shown. (**D**) Visualization and validation of the differentially spliced genes in control and *Srsf10* depleted spermatogonia, and the SRSF10-binding peaks are shown. Tracks from integrative genomics viewer and the SRSF10-binding peaks for selected candidate genes (left). Differentially spliced exons are shaded. Gels images show that RT-PCR analysis of AS patterns of the changed splicing genes in control and *Srsf10*-cKO spermatogonia. The specific primers for RT-PCR validation were supplied in **Supplementary file 2**. Schematics of AS events are shown (blue and yellow rectangles) (right). Changes in 'percent spliced in (PSI)' between control and *Srsf10*-cKO spermatogonia are shown below splicing schematics (ΔPSI). SE, skipped exon; A3SS, alternative 3' splice sites. (**E**) The protein level of NASP, BCLAF1, DAZL, and KAT7 was determined in control and *Srsf10*-cKO THY1⁺ spermatogonia by western blot. β-actin and β-tubulin were used as the internal control.

The online version of this article includes the following source data and figure supplement(s) for figure 8:

**Source data 1.** Differential alternative splicing events in the Iso-seq data using SUPPA2 analysis and RNA-seq data using comprehensive AS Hunting analysis and raw GO analysis data of all 675 differential alternative splicing genes.

**Source data 2.** Raw GO terms for 113 genes directly bound by SRSF10 with significantly affected AS events.

**Source data 3.** Raw RT-PCR gel for *Bclaf1*, *Nasp*, *Sycp1*, *Cenph*, *Rif1,* and *Mcm10*.

**Source data 4.** Raw western blot for BCLAF1, NASP, DAZL, and KAT7.

**Figure supplement 1.** Comparison between RNA-seq and linear amplification of complementary DNA ends and sequencing (LACE-seq) data of THY1⁺ spermatogonia enriched from control and *Srsf10*-cKO testes at postnatal day 6 (P6).

**Figure supplement 1—source data 1.** Lists of the differential expression genes in isolated THY1⁺ spermatogonia in control and *Srsf10*-cKO at P6.

**Figure supplement 2.** The differentially alternative splicing (AS) analysis of isoform-sequencing and RNA-seq data.

**Figure supplement 3.** Visualization and validation of some differentially spliced functional genes in control- and *Srsf10*-depleted spermatogonia.

**Figure supplement 3—source data 1.** Raw RT-PCR gel for *Dazl*, *Kat7*, *Zfp207*, *Exo1*, *Ret*, *Ccna2*, *Ccne2*, and *Cdc7*.

**Figure supplement 3—source data 2.** Relative expression of *Kit* in control and *Srsf10*-cKO spermatogonia.

We then asked how these differentially spliced genes were regulated by SRSF10. Among the 675 differentially spliced genes, 113 genes were bound by SRSF10 as identified in LACE-seq (**Figure 8B**), indicating that these genes are probably the direct targets of SRSF10. The 113 genes were involved in RNA splicing, regulation of translation, chromosome segregation, synaptonemal complex assembly, mitotic spindle organization, and regulation of double-strand break repair (**Figure 8C**). Importantly, we successfully verified that loss of SRSF10 in spermatogonia resulted in differential splicing of exon 7 in *Nasp* (increased exon skipping), exon 6 in *Sycp1* (increased exon skipping), alternative 3' splice sites of exon 5 in *Bclaf1*, exon 32 in *Rif1* (increased exon skipping), exon 8 in *Cenph* (increased exon skipping), and exon 5 in *Mcm10* (increased exon skipping) (**Figure 8D**). The remaining 562 differentially spliced genes were not detectably bound by SRSF10 (**Figure 8B**), suggesting their altered splicing might be an indirect effect of SRSF10 depletion, for example, a secondary result of altered splicing of other splicing factors (e.g. RBM5, RBM19, and HNRNPC) in the absence of SRSF10 (**Figure 8C**). Some genes involved in important spermatogonia-related processes, including cell cycle and germ cell development were also successfully verified. For example, loss of *Srsf10* resulted in differential splicing of exon 8 in *Dazl* (increased exon inclusion), alternative last exon in *Ret*, alternative 5' splice sites of exon 9 in *Kit* (absence of four amino acids), exon 8 in *Cdc7* (increased exon inclusion), exon 9 in *Zfp207* (increased exon skipping), exon 14 in *Kat7* (increased exon skipping), exon 7 in *Ccna* (increased exon skipping), alternative first exon in *Ccne2*, and alternative 5' splice sites of exon 10 in *Exo1* (**Figure 8—figure supplement 3**). Furthermore, we confirmed the corresponding changes at the protein level of the direct targets (*Nasp* and *Bclaf1*) and indirect targets (*Dazl* and *Kat7*) (**Figure 8E**). These data suggest that SRSF10 is required for the correct splicing of functional genes, thus maintaining the homeostasis of AS during spermatogenesis.

## Discussion

The foundation of spermatogenesis is SSCs which can self-renew to maintain the stem cell pool or undergo differentiation into spermatogonial progenitors that expand and further differentiate. Therefore, fully understanding the regulation of proliferation and differentiation of SSCs is of great importance. Recently, the importance of AS as a post-transcriptional regulatory mechanism involved in spermatogonia development is just beginning to be unraveled (**Schmid et al., 2013**; **Elliott and Grellscheid, 2006**). However, the knowledge is still very limited regarding how the splicing machinery, including specific splicing factors that are involved in this process. By illustrating the function and

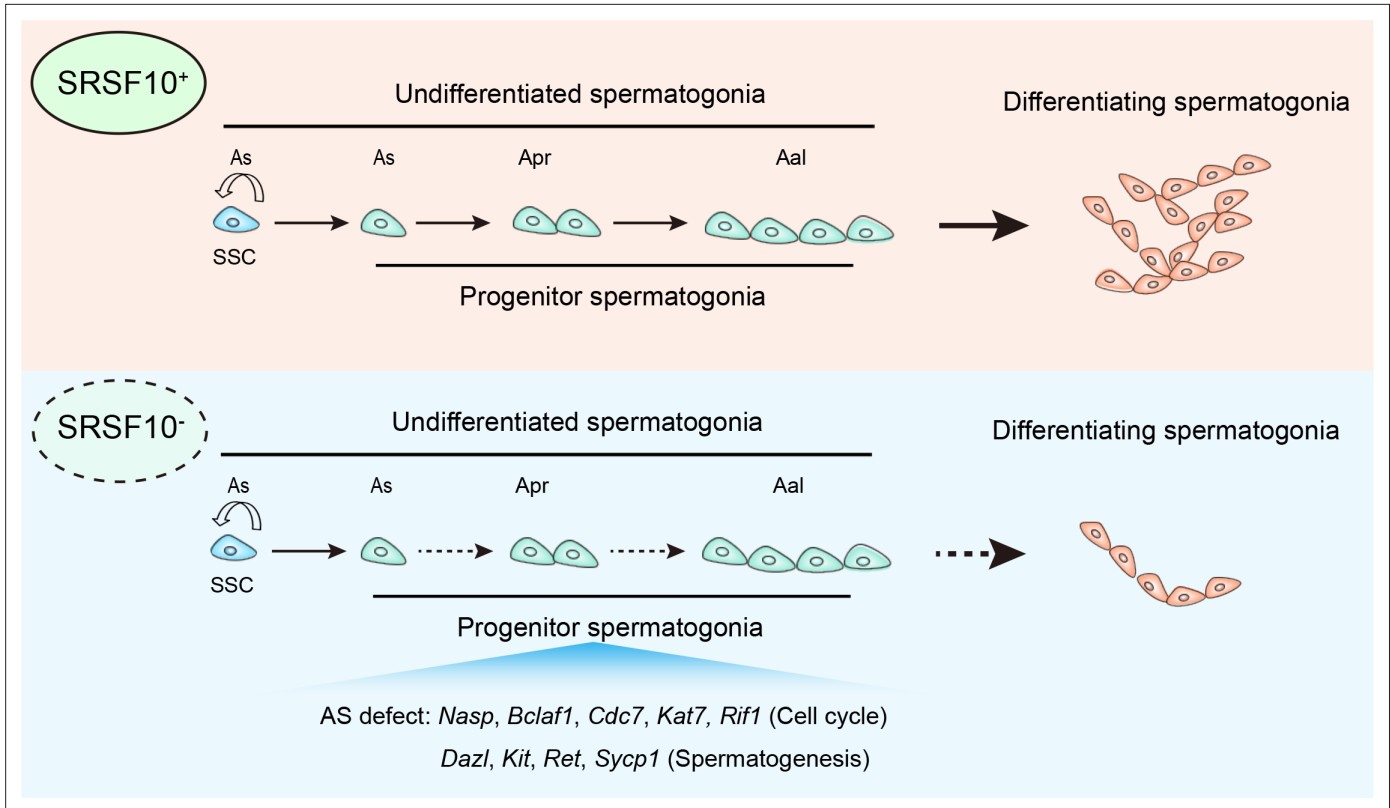

**Figure 9.** Model of SRSF10-mediated alternative splicing (AS) regulation in the development of undifferentiated spermatogonia. In the control testes, SRSF10 promotes the expansion of progenitor spermatogonia and subsequent differentiation of spermatogonia. In the absence of SRSF10, the AS defects of key genes involved in cell cycle, chromosome segregation, and spermatogenesis impair the expansion of progenitor spermatogonia, leading to the decreased number of undifferentiated spermatogonia and nearly absence of differentiating spermatogonia.

mechanism of *Srsf10* in mouse spermatogenesis (**Figure 9**), our study highlights the complexity and requirements of AS regulation in spermatogonia expansion and meiosis initiation.

## Depletion of *Srsf10* leads to inefficient spermatogonia expansion

In adult testes, *Srsf10* depletion mediated by Vasa-Cre leads to the absence of differentiating spermatogonia, with the existence of only fewer PLZF[+] undifferentiated spermatogonia in adult testes, suggesting that the undifferentiated spermatogonia cannot undergo efficient expansion during SSC-dependent spermatogenesis. In the neonatal mice testes, the number of PLZF[+] undifferentiated spermatogonia increases but at a much slower pace from P6 to P12, and differentiating spermatogonia and spermatocytes can barely be observed. This phenotype is much earlier and more severe than that of the depletion of genes that are important for the maintenance of SSCs (*Etv5*, *Bcl6b*, *Id4*, and *Taf4b*; *Oatley et al., 2006*; *Oatley et al., 2011*; *Chen et al., 2005*; *Falender et al., 2005*) or differentiation of spermatogonia (*Sohlh1* and *Sohlh2*; *Hao et al., 2008*; *Ballow et al., 2006*). The former does not lead to infertility until adulthood and exhibits an age-dependent progressive loss of germ cells, while the latter leads to loss of differentiating spermatogonia but with limited effect on undifferentiated spermatogonia. So, the defects of *Srsf10*-cKO testes might mainly be due to failed self-renewal of SSCs and expansion of the progenitor spermatogonia population.

The number of GFRα1[+] cells was decreased in P6 *Srsf10*-cKO testes, and SSC markers expression was not upregulated from P3 to P8, suggesting that loss of SRSF10 in spermatogonia might not cause the accumulation of SSCs. Although the ratio of USPG1 (SSCs) was increased in *Srsf10*-cKO samples as detected in the scRNA-seq data, it is more likely a result of the loss of progenitors (USPG2) and differentiating cells (DSPG1 and DSPG2).

scRNA-seq data showed that the expression of genes associated with the cell cycle in USPG1 (corresponding to SSCs) and USPG2 (corresponding to progenitor cells) was abnormal in *Srsf10*-cKO

testes. However, EdU incorporation was reduced in PLZF[+] spermatogonia in P6 *Srsf10*-cKO testes but was comparable between control and *Srsf10*-cKO GFRα1[+] cells in P3 testes, suggesting that the cell cycle defect was more severe in the progenitor cells than in SSCs. This might be due to the faster proliferation rate of progenitor spermatogonia than SSCs which are normally quiescent and exhibit a very slow cell cycle rate (*de Rooij and Russell, 2000*). Additionally, the PLZF[+] spermatogonia gradually increased, but at a slower pace, in *Srsf10*-cKO testes from P3 to P12, and the expression of genes involved in the expansion and early differentiation of progenitor spermatogonia was also somewhat upregulated from P6 to P8 *Srsf10*-cKO testes. Although we cannot fully exclude the possibility that the cell cycle defect in SSCs impedes the transition to progenitors, the significantly decreased undifferentiated spermatogonia is more likely a result of abnormal cell cycle or proliferation of progenitor cells.

## Depletion of *Srsf10* leads to failure of the first round of spermatogenesis

In neonatal mice, the first round of spermatogenesis initiates directly from prospermatogonia to KIT[+] differentiating spermatogonia, which continue to develop and produce the first wave of spermatozoa (*Yoshida et al., 2006*; *Law et al., 2019*). Meanwhile, another set of prospermatogonia become SSCs which are capable of self-renewal or differentiate into progenitors to initiate spermatogenesis and support the continuity of steady-state spermatogenesis (*Yoshida et al., 2006*). In P3 *Srsf10*-cKO testes, despite the drastic loss of SRSF10 protein, the number of MVH[+] germ cells was comparable to that in control, suggesting that prospermatogonia can probably develop into KIT[+] differentiating spermatogonia. However, the meiosis initiation was still impaired in the P8-P15 *Srsf10*-cKO testes during the first wave of spermatogenesis. This suggests that loss of SRSF10 might lead to developmental arrest of KIT[+] differentiating spermatogonia or SRSF10 might be also involved in the initiation of meiosis. In 2-month-old *Srsf10*-cKO testes, differentiating spermatogonia cannot be detected, indicating that the inefficient expansion of PLZF[+] undifferentiated spermatogonia may lead to failure of differentiation during the subsequent rounds of SSC-dependent spermatogenesis.

## A unique cluster of cells in *Srsf10*-cKO spermatogonia

In the absence of SRSF10, it is interesting that a unique cluster of cells, the DSPG3 was only observed in the scRNA-seq data of the *Srsf10*-cKO group. These cells express the differentiated markers *Stra8*, *Kit*, and *Sohlh1*, but at a lower level than that in the DSPG1 and DSPG2. We speculate that these cells might be directly transited from the fetal prospermatogonia in the first round of spermatogonia (*Yoshida et al., 2006*), or a population of abnormally differentiated cells in the process of spermatogonia differentiation, but later cell cycle defects of these cells lead to developmental arrest.

## Srsf10 depletion leads to extensive AS alterations of functional genes in mouse spermatogonia

As a splicing factor, SRSF10 depletion in spermatogonia affected the splicing of several hundreds of genes, among which were 113 genes that were directly bound by SRSF10 as detected by LACE-seq. These targets are involved in many fundamental biological processes, such as regulation of double-strand break repair, chromosome segregation, and mitotic spindle organization. Importantly, aberrant splicing of some functional genes can be successfully verified experimentally at both the mRNA and protein levels. For example, *Nasp*, which encodes a histone chaperone that preferentially transports H1 histones into nuclei and exchanges H1 histones with DNA (*Richardson et al., 2006*), preferentially skips exon 7 in the absence of SRSF10 in the spermatogonia. This leads to a dramatic decrease of the long isoform (also known as testicular NASP or tNASP; *Richardson et al., 2000*) and an increase of a shorter isoform. Notably, depletion of tNASP reportedly inhibits proliferation in prostate cancer PC-3 cells and renal cell carcinoma cells (*Alekseev et al., 2011*; *Fang et al., 2015*), echoing the cell cycle defects of *Srsf10*-cKO spermatogonia. *Bclaf1* is a known target of SRSF10, and the inclusion of *Bclaf1* exon5a was directly regulated by SRSF10 (*Zhou et al., 2014a*). In colorectal cancer cells, knockdown of the BCLAF1-L isoform (exon 5 a included) inhibited growth, and overexpression of the BCLAF1-L isoform increased tumorigenic potential (*Zhou et al., 2014a*). In *Srsf10*-cKO-enriched spermatogonia, the inclusion of *Bclaf1* exon5a was decreased at both the RNA and protein levels, leading to downregulation of BCLAF1-L isoform and thus proliferation defects. Therefore, the negative effects on cell

proliferation of *Srsf10*-depleted spermatogonia might be due to abnormal AS of abundant functional genes (*Figure 8*). However, how the aberrant splicing of these genes affects their functions in spermatogonia needs to be further clarified.

Besides the factors that are involved in fundamental cellular events, we found and verified the aberrant splicing of many genes that play critical roles in male germ cells, including *Dazl*, *Kit*, *Ret*, and *Sycp1*, in *Srsf10*-deficient spermatogonia (*Figure 9*). Although the functions of specific isoforms and the effects of isoform switches in these genes in spermatogonia are unclear at the moment, specific targets or functions of different isoforms are reported in other cell lines (*Yoshinaga et al., 1991*; *Caruana et al., 1999*; *Crosier et al., 1993*; *Hofmann, 2008*; *Naughton et al., 2006*; *Parker et al., 2014*). For example, the specific mRNA targets for both DAZL isoforms (the exon 8-included long isoform is increased in *Srsf10*-cKO testes) were identified in ESCs, implying the potential functional divergence of these two isoforms in spermatogenesis (*Xu et al., 2013*). We speculate that the splicing of these genes should be precisely regulated to keep the balance of their isoforms and exert normal functions in spermatogenesis. However, it should be noted that these genes are not directly bound by SRSF10, indicating that their abnormal splicing may be a secondary effect of *Srsf10* depletion.

Interestingly, besides changes in gene splicing as expected, depletion of *Srsf10* in spermatogonia leads to changes in gene expression, especially when lower the cutoff (378 upregulated and 206 downregulated). However, almost none of these genes were the direct targets of SRSF10 (24 genes in 1207 genes, 0.041% overlap), and nearly no DSEs were observed (27 genes in 675 genes, 4% overlap). Considering the altered cell types (e.g. the loss of progenitors and differentiating spermatogonia) and gene splicing in the absence of SRSF10, changes in gene expression might be regulated indirectly by SRSF10.

## Materials and methods
### Mice
All mice were maintained under specific-pathogen-free conditions and the illumination time, temperature, and humidity were all in accord with the guidelines of the Institutional Animal Care and Use Committee of the Institute of Zoology at the Chinese Academy of Sciences (CAS; SYXK 2018–0021). The procedures for care and use of animals were approved by the Ethics Committee of the Institute of Zoology at the Chinese Academy of Sciences (IOZ20180083). The *Srsf10*^Floxed/Floxed^ (*Srsf10*^F/F^) mice were purchased from Shanghai Model Organisms Center, Inc. The *Srsf10*^F/F^ male mice were mated with *Vasa-Cre* transgenic females to obtain the mouse model with *Srsf10* specific deletion in the germ cell lines. The purchased *Srsf10*^F/F^ mice are on C57BL/6 J genetic background, while the Vasa-Cre mice are on a mixed background. *Srsf10*^F/+^;Vasa-Cre males (the floxed allele in mature sperm will be deleted by Vasa-Cre and thus generate a knockout allele ['-']), and *Srsf10*^F/F^ females were used for breeding. Four genotypes in the progeny, including *Srsf10*^F/+^, *Srsf10*^F/−^, *Srsf10*^F/+^;*Vasa-Cre*, and *Srsf10*^F/−^;*Vasa-Cre* can be identified. The genotype of *Srsf10*^F/−^;*Vasa-Cre* was used as mutants and was referred to as *Srsf10*-cKO. The genotypes of *Srsf10*^F/+^;*Vasa-Cre* were used as control. Genotyping of *Srsf10* was performed by PCR of mice tail genomic DNA. Forward primer: 5-AACATTTAGCAC ATTTGAGGAT-3, and reverse primer: 5-AACAGCCATATTAACCCGTCTTG-3 were used to detect the wild-type allele (615 bp) and the floxed allele (467 bp). Forward primer: 5-AGCATGCCTATCTTGT GT-3, and reverse primer: 5-TAACCCGTCTTGTAGTAAATCT-3 were used to detect the mutant allele (300 bp). The *Vasa-Cre* was genotyped with forward primer (5-CACGTGCAGCCGTTTAAGCCGCGT -3) and reverse primer (5-TTCCCATTCTAAACAACACCCTGAA-3) with the product was 240 bp. The annealing temperature of all primers was 58°C.

### Fertility test of male mice
Each control and *Srsf10*-cKO male mice (8–12 weeks, n=8) were caged with two 8-week-old wild-type ICR females, and vaginal plugs were checked every morning. Once a vaginal plug was identified (day 1 postcoitus), the male was allowed to rest for 2 days. Two days later, another two females were placed in the cage for another round of mating until two to three plugged females for each male mouse were obtained. The plugged females were separated and monitored for their pregnancy. The fertility test lasted for at least 4 weeks.

## Histological analysis, immunostaining, and imaging

For histological analysis, testes from control and *Srsf10*-cKO were isolated and fixed in Bouin's solution (saturated picric acid: 37% formaldehyde:glacial acetic acid = 15:5:1) overnight at room temperature (RT). These testes were dehydrated through a graded ethanol (30, 50, 70, 80, 90, 100, and 100%) 1 hr for each step, xylene three times, 1 hr for each, paraffin (65°C), two changes, 2 hr each, and embedded testes in paraffin. Then, paraffin-embedded samples were cut into sections of 5 μm thickness. After dewaxing and hydration, the sections were stained with hematoxylin and 1% eosin and imaged with a Nikon ECLIPSE Ti microscope.

For immunostaining, testes from control and *Srsf10*-cKO mice were isolated and fixed in 4% PFA overnight at 4°C. Following dehydration, the testes were embedded in paraffin and cut into sections of 5 μm thickness. After dewaxing and hydration, the sections were boiled in citrate antigen retrieval solution (0.01 M citric acid/sodium citrate, pH 6.0) for 20 mins in the microwave oven. After free cooling, the sections were washed with PBS (pH 7.4) three times and blocked with 5% BSA in PBS for 1 hr at RT. Then, the sections were incubated with primary antibody diluted with 1% BSA overnight at 4°C. On the second day, we washed the sections with PBS three times and incubated them with the secondary antibody diluted with 1% BSA for 1 hr at RT.

After washing in PBS three times, the sections were incubated with 2 μg/ml of Hoechst 33342 (Sigma, B2261) diluted with PBS for 15 min at RT. Finally, the sections were washed with PBS two times and mounted with Fluoromount-G medium (Southern Biotech, 0100–01). The immunofluorescence staining was imaged with a laser scanning confocal microscope LSM880 (Carl Zeiss, Germany).

The primary antibodies used were listed as follows: rabbit anti-SRSF10 polyclonal antibody (ab254935, Abcam, 1:200); goat anti-PLZF polyclonal antibody (AF2944, R&D, 1:200); goat anti-KIT polyclonal antibody (AF1356, R&D, 1:200); goat anti-GFRα1 polyclonal antibody (AF560, R&D, 1:200); rabbit anti-DDX4/MVH polyclonal antibody (ab13840, Abcam, 1:200); rabbit anti-phospho-Histone H2A.X (Ser139/Tyr142) antibody (#5438, Cell Signaling Technology, 1:200); mouse anti-SCP3 antibody (mouse monoclonal) (ab97672, Abcam, 1:200); rabbit anti-Cleaved Caspase-3 (Asp175) antibody (#9661, Cell Signaling Technology, 1:200); and rabbit anti-Ki67 polyclonal antibody (ab15580, Abcam, 1:200).

The secondary antibodies used were listed as follows: Alexa Fluor 488 donkey anti-rabbit (Jackson, 1:500); Alexa Fluor 549 donkey anti-rabbit (Jackson, 1:500); Alexa Fluor 488 donkey anti-mouse (Jackson, 1:500); Alexa Fluor 549 donkey anti-mouse (Jackson, 1:500); and Alexa Fluor 488 donkey anti-goat (Jackson, 1:500).

## RNA extraction and qRT-PCR

Total RNA was extracted from whole testes or enriched cells using RNAzol RT (Molecular Research Center. Inc, RN 190) following the manufacturer's instructions. After removing the residual genomic DNA with the DNase I Kit (Promega, M6101), 500 ng of total RNA was reverse-transcribed into cDNAs using the PrimeScript RT Reagent Kit (TaKaRa, RR037A) according to the manufacturer's protocol. qRT-PCR was performed using an Eva Green 2× qPCR MasterMix-No Dye kit (Abm, MasterMix-S) on a LightCycler 480 instrument (Roche). Relative gene expression was analyzed based on the $2^{-\Delta\Delta Ct}$ method with *β-actin* as internal controls. At least three independent experiments were analyzed. All primers were listed in ***Supplementary file 1***.

## Western blot

The protein from testes or enriched cells was extracted using Radio-Immunoprecipitation Assay (RIPA) lysis buffer (50 mM Tris–HCl (pH 7.5), 150 mM NaCl, 1% sodium deoxycholate, 1% Triton X-100, 0.1% Sodium dodecyl sulfate (SDS), 5 mM EDTA, 1 mM $Na_3VO_4$, and 5–10 mM NaF) containing a protease inhibitor cocktail (Roche, 04693132001), and the protein lysis buffers were incubated on ice for 20 min. After the ultrasound, the protein lysis buffers were centrifuged at 4°C, 12,000 rpm for 20 min, and quantified using a BCA reagent kit (Beyotime, P0012-1). After being boiled at 95°C for 6 min, equal amounts of total protein lysates were used for immunoblotting analysis. Different molecular weight proteins were separated in a 10% SDS–PAGE gel and transferred onto polyvinylidene fluoride (PVDF) membranes. After blocking with 5% non-fat milk dissolved with Tris Buffered Saline with Tween 20 (TBST, pH 7.4) for 1 hr at RT, the membranes were incubated with diluted primary antibodies diluted with 1% BSA at 4°C overnight. After three washes with TBST, the membranes were

incubated with secondary antibodies diluted with TBST at RT for 1 hr. The signals were developed with Pierce ECL Substrate (Thermo Fisher Scientific, #34080), detected with Bio-RAD ChemiDocTMXRs+, and analyzed with Quantity One software (Bio-Rad Laboratories). All primary antibodies were listed as follows, rabbit anti-SRSF10 polyclonal antibody (NB110-93598, Novus Biologicals, 1:1,000); goat anti-PLZF polyclonal antibody (AF2944, R&D, 1:1,000); rabbit anti-DDX4/MVH polyclonal antibody (ab13840, Abcam, 1:1,000); rabbit anti-phospho-Histone H2A.X (Ser139/Tyr142) antibody (#5438, Cell Signaling Technology, 1:1,000); mouse anti-human DAZL monoclonal antibody (MCA23336, Bio-Rad, 1:500); BCLAF1 rabbit polyclonal antibody (26809–1-AP, Proteintech, 1:1000); NASP rabbit polyclonal antibody (11323–1-AP, Proteintech, 1:1000); and KAT7 rabbit polyclonal antibody (13751–1-AP, Proteintech, 1:1000).

The secondary antibodies used were listed as follows: Goat Anti-Rabbit IgG (H+L), HRP Conjugated (EASYBIO, BE0101, 1:3000); Goat Anti-Mouse IgG (H+L)-HRP Conjugated (EASYBIO, BE0102, 1:3000); Rabbit Anti-Goat IgG (H+L)-HRP Conjugated (ZSGB-BIO, ZB-2306, 1:3000).

## Whole-mount immunostaining

The testes were collected and dissected to remove the tunica albuginea. Seminiferous tubules were dispersed with tweezers and fixed at 4% PFA overnight at 4 °C. The tubules were washed three times with PBST (pH 7.4) for 10 min each and permeated with 0.1% TritonX-100 for 4 h at RT. Then, the tubules were washed three times with PBST and blocked with 5% BSA in PBST for 2 h at RT. The tubules were incubated with primary antibodies diluted with 1% BSA at 4 °C overnight. After washing with PBST and the tubules were incubated with secondary antibodies diluted with PBST for 4 h at RT. With the washing of PBST, the tubules were stained with Hoechst diluted with PBST for 30 min at RT. After simply washing, the tubules were mounted with Fluoromount-G medium (Southern Biotech, 0100–01). The imaging of whole-mount staining followed the protocol described above.

## RNA sequencing

Testes samples were collected from P3, P6, and P8 control and *Srsf10*-cKO mice. THY1+ spermatogonia were isolated from P6 control and *Srsf10*-cKO mice. The RNA-seq experiment was performed in two biological replications of P3, P6, and P8 testes and isolated THY1+ spermatogonia in control and *Srsf10*-cKO mice. Total RNA was isolated using the RNAzol RT (Molecular Research Center. Inc, RN 190) according to the manufacturer's protocol and treated with DNase I to remove residual genomic DNA. A total amount of 1 µg of RNA per sample was used to prepare cDNA libraries generated using the NEBNext Ultra RNA Library Prep Kit for Illumina (NEB) following the manufacturer's instructions. 6 G base pairs (raw data) were generated by Illumina Novaseq 6000 for each cDNA library. The adaptor sequence and sequences with a high content of unknown bases or low-quality reads were removed to produce the clean reads used for bioinformatic analysis.

## RNA-seq bioinformatic analyses

After initial quality control, the clean reads were aligned to the mouse reference genome (mm 9) using Tophat v2.1.1 (*Trapnell et al., 2009*). Next, the gene expression was calculated by Cufflinks v2.2.1. The normalization of gene expression values was based on theFPKM.

## EdU incorporation assay

EdU (RiboBio, C00053) dissolved in PBS was injected intraperitoneally at 5 mg/kg of body weight. Testes were collected from control and *Srsf10*-cKO mice 4 hr later following EdU incorporation. For the EdU and PLZF co-staining, the testes were fixed in 4% PFA overnight at 4°C, embedded in paraffin, and then performed according to the protocol of Cell-Light Apollo567 Stain Kit (100T) (RiboBio, C10310-1). Briefly, paraffin-embedded samples were cut into sections of 5 µm thickness. After dewaxing and hydration, the sections were washed with 2 mg/ml glycine for 10 min, incubated with 0.5% TrionX-100 in PBS for 10 min, and then washed with PBS for 5 min. Next, the section was stained with fresh 1×Apollo reaction solution for 30 min at RT. The section was washed with 0.5% TrionX-100 three times for 10 min each time, washed with methanol two times for 5 min each time, and PBS two times for 5 min each. Then, the sections were blocked with 5% BSA in PBS for 1 hr at RT, incubated with primary antibody diluted with 1% BSA overnight at 4 °C. On the second day, we washed the sections with PBS three times and incubated them with secondary antibody diluted with

1% BSA for 1 hr at RT. After washing in PBS three times, the sections were incubated with 2 µg/ml of Hoechst 33342 (Sigma, B2261) diluted with PBS for 15 min at RT. Finally, the sections were washed with PBS two times and mounted with Fluoromount-G medium (Southern Biotech, 0100–01). The immunofluorescence staining was imaged with a laser scanning confocal microscope LSM880 (Carl Zeiss, Germany).

## Testes digestion and generation of cell suspensions

Testes from P6 mice were used to generate single-cell suspensions following enzymatic digestion as described previously (*Oatley and Brinster, 2006*). In brief, testes were collected from P6 or P8 control and *Srsf10*-cKO mice in Hanks' balanced salt solution, and the tunica albuginea was removed. Then, the testes were digested with 4.5 ml 0.25% trypsin/EDTA and 0.5 ml 7 mg/ml DNase I solution (Sigma, d5025) for 6 min at 37°C. After gently pipetting, another 0.5 ml DNase I solution was added to the cell suspension to digest for another 2–3 min at 37°C and followed by the addition of 1 ml 10% FBS. Single-cell suspensions were made by gently repeated pipetting and passed through a 40-µm pore size cell strainer. The cells are centrifugated at 600 g for 7 min at 4°C and resuspended with 180 µl running buffer (Miltenyi Biotec, 130-091-221) for MACS selection.

## THY1$^+$ spermatogonia isolation

We added 20 µl THY1 antibody-conjugated microbeads (anti-mouse CD90.2 [Thy1.2] MicroBeads; Miltenyi Biotech, 130-121-278) to the single-cell suspensions and mixed well, then incubated for 20 min at 4°C. After incubation, 2 ml running buffer was added to the cells and centrifugated at 600 g for 7 min at 4°C. The pellet was then resuspended in 500 µl of running buffer for magnetic cell separation. The LS columns (Miltenyi Biotec, 130-042-401) were placed in the magnetic field of the MACS Separator and prewashed with 0.5 ml of running buffer followed by the addition of the THY1-labeled cell suspension. Wash the LS columns three times with 3 ml running buffer, and the unlabeled THY1$^-$ cells were eluted. Then, the LS columns were removed from the separator and placed on new 15 ml centrifuge tubes. The THY1$^+$ cells were eluted from the LS columns in a 5-ml running buffer and centrifuged at 600 g for 7 min at 4°C. After centrifugation, the supernatant is removed, and the pellet of THY1$^+$ cells was stored at –80°C for subsequent sequencing analysis and validation.

## Droplet-based single-cell RNA sequencing

THY1$^+$ KIT$^-$, KIT$^+$ THY1$^-$, and THY1$^+$KIT$^+$ spermatogonia collection: Testes collected from P8 control and *Srsf10*-cKO mice were digested, and the single-cell suspensions were generated following the protocol described above. We added 20 µl THY1 antibody-conjugated microbeads (anti-mouse CD90 [Thy1.2] MicroBeads; Miltenyi Biotech, 130-121-278) and 40-µl KIT antibody-conjugated microbeads (anti-mouse CD117 MicroBeads; Miltenyi Biotech, 130-091-224) to the single-cell suspensions for 20 min at 4°C, and followed the protocol described above to get the THY1$^+$ KIT$^-$, KIT$^+$ THY1$^-$, and THY1$^+$KIT$^+$ spermatogonia.

The single-cell RNA-seq experiment was performed with one replicate of control that was collected from three mice, and two replications of *Srsf10*-cKO that were collected from six mice, respectively. Single cells were suspended in PBS containing 0.04% BSA and loaded onto the Chromium 3' v3 platform (10× Genomics) to generate single-cell libraries according to the manufacturer's protocol. Briefly, single cells were partitioned into Gel Bead-In-EMulsions (GEMs) in the 10× Chromium Controller instrument followed by cell lysis and barcoded reverse transcription using a unique molecular identifier (UMI). The cDNA was generated and then amplified, and the libraries were finally sequenced using an Illumina Novaseq 6000 sequencer with a sequencing depth of at least 100,000 reads per cell with a pair-end 150 bp (PE150) reading strategy (performed by CapitalBio Technology, Beijing).

## Data processing of scRNA-seq

Raw data were processed using the Cell Ranger Software (version 4.0.0). Briefly, the raw files were converted to demultiplexed fastq files through the Cell Ranger mkfastq pipeline. The fastq files were then aligned to the mouse reference genome (mm 10) using the STAR aligner. Next, the reads were further filtered for valid cell barcodes and UMIs to produce a count matrix. Finally, the count matrix was imported into the R package Seurat, and quality control was performed to remove outlier cells and genes. Cells with 200–4000 detected genes were retained. Genes were retained in the data if

they were expressed in ≥3 cells. After applying these quality control criteria, 1923 cells and 23,310 genes remained for further analysis. Additional normalization was performed in Seurat on the filtered matrix to obtain the normalized count. Highly variable genes across single cells were identified, and principal components analysis was performed to reduce the dimensionality. Then the top 16 principal components were used for cluster/subtype identification and UMAP visualization (*Becht et al., 2018*). Known cell type-specific markers were used to match each cell type.

## Pseudotime analysis of single-cell transcriptomes

Germ cell lineage trajectories were constructed according to the procedure recommended in the Monocle3 documentation (*Trapnell et al., 2014*, https://cole-trapnell-lab.github.io/monocle3/docs/starting/). Briefly, the top differentially expressed genes were selected as 'ordering genes' to recover lineage trajectories in Monocle3 using default parameters. After pseudotime was determined, differentially expressed genes were clustered to verify the fidelity of lineage trajectories.

## Linear amplification of complementary DNA ends and sequencing

SRSF10 LACE-seq was performed based on a previously described protocol with some modifications. Briefly, THY1$^+$ cells were collected into 1.5 ml LoBind microcentrifuge tubes (Eppendorf, 022431021) with 5 µl of 1×PBS (pH 7.4). The cells were irradiated twice with 0.40 J cm$^{-2}$ UV light (254 nm) in a CL-1000 ultraviolet crosslinker (UVP). 10 µl of protein A/G magnetic beads (per sample, Thermo Scientific, 26162) were washed twice with BSA/PBS solution (0.1% BSA in 1×PBS) and incubated with 200 µl blocking buffer (1×PBS, 0.2 mg/ml glycogen, 0.2 mg/ml BSA) at RT for 1 hr. The blocked beads were then washed once with 0.1 M Na-phosphate buffer (93.2 mM Na$_2$HPO$_4$, 6.8 mM NaH$_2$PO$_4$, 0.05% tween 20, pH 8.0) and then resuspended in 40 µl of 0.1 M Na-phosphate buffer containing 5 µg SRSF10 antibody (Cat No.) or IgG (Cat No.), incubating at RT for 1 hr. The antibody-coupled beads were washed twice with wash buffer (1×PBS, 0.1% SDS, 0.5% NP-40, 0.5% sodium deoxycholate) and resuspended in 10 µl of wash buffer per sample. The cross-linked samples were lysed on ice with 50 µl of wash buffer for 10 min. After removal of genomic DNA with 4 µl of RQ1 DNase (Promega, M6101) in the presence of 1 µl of RNase inhibitor (Thermo Scientific, EO0381), 10 µl of antibody-coupled beads were added to the lysate and incubated for 1 hr at 4°C. Bead-bound antibody-RNA complexes were then washed twice with wash buffer, once with high-salt wash buffer (5×PBS, 0.1% SDS, 0.5% NP-40, 0.5% sodium deoxycholate), and once with PNK buffer (50 mM Tris-HCl, pH 7.4, 10 mM MgCl$_2$, 0.5% NP-40). The immunoprecipitated RNAs were then fragmented with 1×10$^{-5}$ U of micrococcal nuclease (MNase, New England BioLabs, M0247S) in 1×MN reaction buffer (50 mM Tris-HCl, pH 8.0, 5 mM CaCl$_2$) for 3 min at 37°C. After another round of extensive washing (twice with 1×PNK + EGTA buffer [50 mM Tris-HCl, pH 7.4, 20 mM EGTA, 0.5% NP-40], twice with wash buffer, and twice with PNK buffer), the RNA 3′ ends were dephosphorylated on beads with FastAP alkaline phosphatase (Thermo Scientific, EF0651) for 15 min in FastAP buffer. After washing beads twice with 1×PNK + EGTA buffer, twice with 1×PNK buffer, and twice with BSA solution (0.2 mg/ml BSA in DEPC water), the 3′ linker was ligated with T4 RNA ligase 2 (New England BioLabs, M0242) for 2.5 hr at RT. After washing beads three times with 1×PNK buffer, the RNA was reverse transcribed with Superscript II reverse transcriptase (Thermo Scientific,18064014) according to the manufacturer's instruction. Then excess T7-RT primer was digested with Exonuclease I (New England BioLabs, M0293), first-strand cDNA was released from Protein A/G beads by treatment with RNase H (Thermo Scientific, EN0202) and captured by streptavidin C1 beads (Thermo Scientific, 650002). After washing beads twice with 1×B & W buffer (5 mM Tris-HCl, pH 7.5, 0.5 mM EDTA, 1 M NaCl), once with BSA solution, the cDNA 3′ linker was ligated with T4 RNA ligase 1 (New England BioLabs, M0437) overnight at RT. Then cDNA was pre-amplified using KAPA HiFi HotStart ReadyMix (KAPA Biosystems, KK2601), and PCR products were purified with Ampure XP beads (Beckman Coulter, A63881). Then in vitro transcription was performed, the DNA template was removed with TURBO DNase (Thermo Scientific, AM2238), and the RNA was purified with Agencourt RNA Clean beads (Beckman Coulter, A63987). After reverse transcription and indexed PCR, the PCR products were size-selected on a 2% agarose gel, and regions corresponding to 250–500 bp were purified using Gel Extraction Kit (Qiagen, 28604). The LACE-seq library was paired-end sequenced using Illumina NovaSeq 6000 at Novogene.

## LACE-seq data processing

The first 4 nt at 5' end of read1 and read2 were extracted as UMI and then the adaptor sequences, poly(A) tails at the 3' end and low-quality reads were removed using fastp (*Chen et al., 2018*) with parameters '--detect_adapter_for_pe -m -e 20 l 18 -y -Y 10 c --overlap_len_require 10 --overlap_ diff_percent_limit 10 U --umi_loc per_read --umi_len 4--poly_x_min_len 5'. Clean reads were then first mapped to mouse pre-rRNA using Bowtie2 software (v.2.1.0; *Langmead and Salzberg, 2012*), and the remaining unmapped reads were aligned to mouse mm9 genome using HISAT2 (*Kim et al., 2015*). PureCLIP (*Krakau et al., 2017*) was used to identify peaks with default parameters. For motif analysis, LACE-seq peaks were firstly extended 20 nt to the 5' upstream, and the overrepresented hexamers were identified using the resultant peaks with Homer *Heinz et al., 2010*. The enrichment of mRNA in each sample was calculated using cufflinks 2.2.1 and expressed as FPKM.

## PacBio isoform sequencing

THY1[+] spermatogonia were isolated from P6 control and *Srsf10*-cKO mice. Total RNA was isolated using the RNAzol RT (Molecular Research Center. Inc, RN 190) according to the manufacturer's protocol and treated with DNase I to remove residual genomic DNA. 5 µg of total RNA per sample was used to prepare the Iso-Seq library using the Clontech SMARTer cDNA synthesis kit and the BluePippin Size Selection System protocol as described by Pacific Biosciences (PN 100-092-800-03).

After running the Iso-Seq pipeline, sequence data were processed using the SMRTlink 5.0 software. A circular consensus sequence was generated from subread BAM files. Additional nucleotide errors in consensus reads were corrected using the Illumina RNAseq data with the software LoRDEC. Then all the consensus reads were aligned to the mouse reference genome (mm 9) using Hisat2.

## Differential splicing analysis and validation

Differential splicing events of Iso-seq data were analyzed using SUPPA2 software with the standard protocol. The SUPPA2 software can identify the seven common modes of AS events and obtain accurate splicing change quantification between control and *Srsf10*-cKO samples. We used $p < 0.05$ and |ΔPSI |>0.1 as the threshold to filter for significantly differential splicing events. Differential splicing events of NGS data were analyzed using CASH software with the standard protocol using $p < 0.05$ as the threshold for filtering. For validation, we first imported the differentially spliced sites of interesting functional genes analyzed by SUPPA2 and CASH into the integrative genomics viewer tool to efficiently and flexibly visualize and explore spliced sites between control and *Srsf10*-cKO samples. The primers for differentially spliced exons were designed using Primer5. Primers were designed within constitutive exons flanking the differentially spliced exons. Standard PCR for analysis by gel electrophoresis was performed according to the manufacturers' instructions and visualized by running on a 2% agarose gel. All primers were listed in *Supplementary file 2*.

## Statistical analysis

All experiments were performed at least three independent times. At least three independent biological samples were collected for the quantitative experiments. Quantification of positively stained cells was counted from at least three independent fields of view. Paired two-tailed Student's t-test was used for statistical analysis, and data were presented as mean ± SEM. *** represent $p < 0.001$, ** represent $p < 0.01$, and * represent $p < 0.05$. $p < 0.05$ was considered with a significant difference level. Equal variances were not formally tested. No statistical method was used to predetermine sample sizes.

## Acknowledgements

We thank Dr. Wei Xie's lab from Tsinghua University for computational facility assistance. We thank Dr. Eugene Yujun Xu from Nanjing Medical University for kindly providing anti-DAZL antibody. We thank Chengpeng Xu, Shiwen Li and Xili Zhu for their technical assistance. Jian Chen and all members of Sun lab for their helpful advice. We thank Mingming Fan for her guidance on MACS of spermatogonia. This study was funded by the National R & D Program (2018YFA0107701), National Natural Science Foundation of China (31801241, 31801240, 81971452, 81871211), the Natural Science Foundation of

Guangdong Province, China (Grant No. 2018A030313665 and 2021 A1515011011), and the Medical Key Discipline of Guangzhou (2021–2023).

## Additional information

### Funding

| Funder | Grant reference number | Author |
|---|---|---|
| National Center for Research and Development | 2018YFA0107701 | Qing-Yuan Sun |
| National Natural Science Foundation of China | 31801241 | Wenbo Liu |
| National Natural Science Foundation of China | 31801240 | Zheng Gao |
| National Natural Science Foundation of China | 81971452 | Jianqiao Liu |
| National Natural Science Foundation of China | 81871211 | Lei Li |
| Natural Science Foundation of Guangdong Province | 2018A030313665 | Wenbo Liu |
| Natural Science Foundation of Guangdong Province | 2021A1515011011 | Zheng Gao |

The funders had no role in study design, data collection and interpretation, or the decision to submit the work for publication.

### Author contributions

Wenbo Liu, Conceptualization, Data curation, Formal analysis, Supervision, Funding acquisition, Validation, Investigation, Visualization, Writing – original draft, Project administration, Writing – review and editing; Xukun Lu, Formal analysis, Visualization, Writing – original draft, Writing – review and editing; Zheng-Hui Zhao, Formal analysis, Validation, Writing – review and editing; Ruibao SU, Qian-Nan Li Li, Yue Xue, Si-Min Sun Sun, Wen-Long Lei, Hanyan Liu, Validation, Investigation; Zheng Gao, Formal analysis, Funding acquisition, Validation; Lei Li, Funding acquisition, Investigation; Geng An, Formal analysis; Zhiming Han, Investigation, Writing – review and editing; Ying-Chun Ouyang, Yi Hou, Investigation; Zhen-Bo Wang, Supervision, Writing – review and editing; Qing-Yuan Sun, Jianqiao Liu, Supervision, Funding acquisition, Writing – review and editing

### Author ORCIDs

Wenbo Liu http://orcid.org/0000-0001-7739-8123
Xukun Lu http://orcid.org/0000-0003-1122-2840
Qing-Yuan Sun http://orcid.org/0000-0002-0148-2414

### Ethics

All mice were maintained under specific-pathogen-free (SPF) conditions and the illumination time, temperature and humidity were all in accord with the guidelines of the Institutional Animal Care and Use Committee of the Institute of Zoology at the Chinese Academy of Sciences (CAS; SYXK 2018-0021). The procedures for the care and use of animals were approved by the Ethics Committee of the Institute of Zoology at the Chinese Academy of Sciences (IOZ20180083).

### Decision letter and Author response

Decision letter https://doi.org/10.7554/eLife.78211.sa1
Author response https://doi.org/10.7554/eLife.78211.sa2

## Additional files

### Supplementary files
- Transparent reporting form
- Supplementary file 1. Primers for quantitative PCR validation.
- Supplementary file 2. Primers for validation of differential splicing changes.

### Data availability

Sequencing data have been deposited in GEO under accession codes GSE190646. Our work did not generate any datasets or use any previously published datasets. Source data for Figures 1, 2, 3, 4, 6, 7 and figure supplement of Figures 1, 3, 7 have been provided.

The following dataset was generated:

| Author(s) | Year | Dataset title | Dataset URL | Database and Identifier |
|-----------|------|---------------|-------------|-------------------------|
| Liu W, Lu X, Zhao Z-H, Liu J | 2022 | SRSF10 is essential for progenitor spermatogonia expansion by regulating alternative splicing | https://www.ncbi.nlm.nih.gov/geo/query/acc.cgi?acc=GSE190646 | NCBI Gene Expression Omnibus, GSE190646 |

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
