## [Editor Report]

The overall conclusion that SRSF10 plays important roles during mouse spermatogenesis, especially during the proliferation of spermatogonial stem cells and meiosis, is supported by the results. Furthermore, the RNA sequencing data could be an excellent resource for other investigators interested in further exploration of the regulation of spermatogenesis by alternative RNA splicing. This manuscript represents a valuable contribution to the literature on the role of alternative splicing in spermatogenesis. It should be of considerable interest to investigators studying post-transcriptional regulation of spermatogenesis, oogenesis, and animal development in general.

---

## [Decision Letter]

**Decision letter after peer review:**

Thank you for submitting your article "SRSF10 is essential for progenitor spermatogonia expansion by regulating alternative splicing" for consideration by *eLife*. Your article has been reviewed by 3 peer reviewers, one of whom is a member of our Board of Reviewing Editors, and the evaluation has been overseen by Mone Zaidi as the Senior Editor. The reviewers have opted to remain anonymous.

Essential revisions:

Overall, the three reviewers agree that the subject of your manuscript could be of interest to readers of *eLife* and that the conclusions are partly supported by the results. However, the reviewers feel that (1) the study is too descriptive and lacks molecular insights, (2) alternative explanations of the results should be considered, (3) there should be more information about whether gene expression changes are really direct targets of SRSF10, (4) there should be additional controls for several of the experiments, and (5) there should be a significant effort to correct a variety of scientific and editorial issues. Consequently, this manuscript should be extensively revised prior to further consideration for publication in *eLife*. The revised version of this manuscript must be submitted within 3 months of receiving this decision letter. In its present form, the manuscript is not appropriate for publication in *eLife*.

Representative comments by the reviewers follow:

The molecular mechanism by which SRSF10 impacts key splicing or gene expression events, and how many of these are direct targets of SRSF10, remains unexplored. The study is highly descriptive and lacks molecular insights. Identification of the molecular changes that are driving the loss of spermatogonia upon depletion of SRSF10 requires (1) demonstrating a change in protein expression or activity that follows the transcript changes and (2) demonstrating that one or more of these changes is sufficient to phenocopy SRSF10-depletion.

Although some of the data showed that the mutant (cKO) mice may contain a deletion mutation of Srsf10 (Figure 1B, Figure 7-Supl 1D), the strategy of generating mutants and genotyping of mice should be presented as data, including the genotyping PCR (gel electrophoresis) of mice and a schematic showing the genomic organization of Srsf10 loci before and after Cre mediated recombination. What does Srsf10F/- represent as described in Materials and methods (Lines 452-453)? The labeling of figures can also be changed from "Control" to "Srsf10F/+; Vasa-Cre" when applicable. Explain what the residual SRSF10 in Figure 7-Supl1D mean for the deletion mutation.

The authors used heterozygous mutants as the control. At least one experiment should be presented to show that the heterozygous mice indeed had same phenotypes as the wild type, and different from the homozygous mutants. This can be done by comparing spermatogenic cells on testis sections or cell sorting of SSCs among the three genotypes.

Maintaining consistency when describing SSC sub-populations may help to avoid confusion. For example, in the Abstract it says "impeded the expansion of progenitor spermatogonia"; in lines 97-99, it "impaired … in undifferentiated spermatogonia"; in lines 234-236, it "may not affect the gene expression and proliferation of SSCs", whereas in Figure 4B, Gfra1 was among the decreased genes at P6. This may be done by clearly defining the particular cellular state with their gene expression signatures. For example, un-differentiated SSCs for cells expressing (GFRa1), undifferentiated progenitors for cells expressing OCT4 and PLZF, differentiating progenitors for cells expressing KIT, cells developed beyond type A1 stage can be referred to as differentiated spermatogonia.

The conclusion that "the progenitor expansion is affected, SSC formation is not" should be drawn carefully, as several of the data suggest otherwise. For example, Figure 3C, using MVH^+^PLZF- as a criterion for selecting differentiated cells may not exclude GFRa1+ SSCs, since the majority of PLZF+ progenitors are not expressing GFRa1. In this connection, Figure 3F, lines 178-179, described the decrease of GFRa1+ cells, which are classified as SSCs in the manuscript. The single cell RNAseq analyses also suggested the changes in the ratios of USSC1 (un-differentiated SSC1, increased from 17.5% to 35.5%) and USSC2 (un-differentiated SSC2, decreased from 20.2% to 8.1%) in mutants, compared to the control. Note that USSC1 expresses high GFRa1 and USSC2 expresses high PLZF and OCT4. It is likely that impeding the progenitors' cell division caused accumulation of SSCs or that SSCs' defect in M phase progression led to decrease of progenitors. This should be discussed in the Discussion section. The authors should also introduce clearly how the single cells' classification on SSCs were done and how they would match the cell types described before RNA sequencing data.

In the Introduction, the rationale of studying SRSF10 in male germ line should be mentioned. In the Discussion, alternative implications and caveats of results should be discussed, including – What does DSSC3 mean for the mutants? What is the reason that DEGs were found in mutants when SRSF10 is a splicing regulator? How is it that genes affected by alternative splicing were not the same as DEGs from RNA sequencing analyses?

The Materials and methods section requires substantial improvements by providing sufficient information on how the experiments were carried out, including details such as: hydration time for fixing testes, companies and catalogs for equipment used, antibodies (mono-/polyclonal) types, solution used for diluting antibodies, pH of PBS used, and how Edu staining was done. Were the scRNAseq done for THY1+KIT+ and THY-/KIT+ cells, were all experiments repeated at least three times including RNA sequencing, etc.

Table-1 describes the average litter size produced by wild type and mutant mice, which is higher than what is usually found for C57BL6/J mice. The number of pups per litter should be from one female mouse, please check and correct. The time period for this recording should also be indicated.

Change "per tubule" to "per cross-section" in Figures when applicable. The number of KIT+ and GFRa1+ cells in Figure 3D and 3F should be quantified (e.g, using average number of cells/per unit length of seminiferous tubule).

In Figure 2A, black arrows indicate apoptotic cells. What criteria were used to tell the difference?

In Figure 2B, state clearly which one is for the control and which one in for the mutant, lines 139-142. Figure 2E is similar to Figure 7-Supl1D. Explain why the mutants contain SRSF10 protein?

In Figure 5E – Use the same scale on X-axis for better comparison.

In Figure 6A – Label top genes in the graph as in Figure 6-Supl2. The number of DEGs described in line 283 is not the same as that shown in Figure 6A. Why is "cellular iron ion homeostasis" not present in up-regulated Gos and "regulation of G2/M transition of mitotic cell cycle" is not present in the down-regulated GOs (line 285) for USSC1 in Figure 6B?

Describe what the gels shown in Figure 7E and corresponding supplements are, make sure the gel images presented are the same as those in the source files, and check if the labeling for each Table is correct in the Source Files.

Define clearly how statistics were set for significances: *, **, ** represent what P values in Materials and methods, as well as in the Figure Legends. Instead of stating "at least 500 tubules or 1000 tubules", present the exact numbers of cross-sections examined for each experiment when applicable in the Figure Legends (e.g., for Figures 2, 3, and 6). Instead of "adults", give the age of mice used for experiments when applicable, either in the Figures or Figure Legends.

Check and correct all grammar and spelling errors throughout the manuscript.

*Reviewer #1 (Recommendations for the authors):*

To some extent the manuscript describes a host of molecular markers (genes) involved in alternative splicing, all of which affect steps in spermatogenesis. Consequently, it is clear that alternative splicing of pre-mRNAs is critical for spermatogenesis, but less clear as to why it is critical. It remains unclear whether or not defects in meiosis initiation by spermatogonal stem cells (SSC) in neonatal mice is due to defects in mitotic progenitor cell proliferation. Is it clear that certain steps in SSC formation are not affected by the deletion of Srsf10? In addition to experiments clearing up this issue, the authors should provide more convincing evidence that they are dealing with completely inactivated Srsf10 knockout mice (e.g., show PCR analysis etc.). Are redundant mechanisms at play here? Of the hundreds of genes in spermatogonia affected by the knockout of Srsf10 (i.e., affected by a lack of alternative splicing), which are thought to be involved in the initiation of meiosis? Are any of these same genes involved in a similar process in females, i.e., the conversion of oogonia (mitotic) into oocytes (meiotic)? The manuscript should undergo major editorial improvements making it more understandable to scientists unfamiliar with spermatogenesis, Srsf10, and alternative splicing (also check the many spelling and grammatical errors).

In general, this could be a valuable contribution to the literature on the role of alternative splicing in spermatogenesis (and possibly oogenesis). However, despite the extensive nature of this manuscript and its potentially important conclusions, it will be necessary for the manuscript to undergo revision prior to acceptance. The revision should include information missing from the manuscript, alternative interpretations of experiments, and editorial improvements (spelling etc.).

*Reviewer #2 (Recommendations for the authors):*

Some main concerns that should be addressed:

1. Although some of the data showed that the mutant (cKO) mice may indeed contain deletion mutation of Srsf10 (Figure 1B, Figure 7-Supl1D), the strategy of generating mutants and genotyping of mice should be presented as data, including the genotyping PCR (gel electrophoresis) of mice, a schematic showing the genomic organization of Srsf10 loci before and after Cre mediated recombination. For the same reason, what Srsf10F/- represents, as described in Materials and methods (Lines 452-453)? The labeling of figures can also be changed from "Control" to "Srsf10F/+;Vasa-Cre" when applicable. Explain what the residual SRSF10 in Figure 7-Supl1D mean for the deletion mutation.

2. The authors used heterozygous mutants as the control. At least one experiment should be presented to show that the heterozygous mice indeed had same phenotypes as the wild type, and different from the homozygous mutants. This can be done by comparing spermatogenic cells on testis sections, or cell sorting of SSCs, among three genotypes.

3. Maintaining consistency when describing SSC sub-populations may help to avoid confusion. For example, in Abstract, "impeded the expansion of progenitor spermatogonia"; in lines 97-99, "impaired … in undifferentiated spermatogonia"; in lines 234-236, "may not affect the gene expression and proliferation of SSCs", whereas in Figure 4B, Gfra1 was among the decreased genes at P6. It may be done by clearly defining the particular cellular state with their gene expression signatures, for example: un-differentiated SSCs for cells expressing (GFRa1), undifferentiated progenitors for cells expressing OCT4 and PLZF, differentiating progenitors for cells expressing KIT, cells developed beyond type A1 stage can be referred to as differentiated spermatogonia.

4. The conclusion of "the progenitor expansion is affected, SSC formation is not" should be drawn carefully, as several data suggest otherwise. For example, Figure 3C, using MVH^+^PLZF- as a criterion for selecting differentiated cells may not exclude GFRa1+ SSCs, since the majority of PLZF+ progenitors are not expressing GFRa1. In fact, Figure 3F, lines 178-179, described the decrease of GFRa1+ cells, which are classified as SSCs in the manuscript. The single cell RNAseq analyses also suggested the changes in the ratios of USSC1 (un-differentiated SSC1, increased from 17.5% to 35.5%) and USSC2 (un-differentiated SSC2, decreased from 20.2% to 8.1%) in mutants, comparing to the control. Note that USSC1 expresses high GFRa1 and USSC2 expresses high PLZF and OCT4. It is likely that impeding progenitors' cell division caused accumulation of SSCs, or SSCs' defect in M phase progression led to decrease of progenies (progenitors). This should be discussed in the Discussion section. The authors should also introduce clearly how the single cells' classification on SSCs were done and how they would match the cell types described before RNA sequencing data.

In the same vein, Figure 3G used MVH to label all germ cells, without considering the SSC sub-groups that may already start to differ at this time point (to support the first wave spermatogenesis), thus this data cannot say about changes in SSC formation. Here, it should be noted that pro-SGs (a.k.a. gnocytes) resume proliferation following the birth of the animal, concomitantly with their migration. The notion described in the text, lines 183-184, "they migrate … then restore proliferation", lines 186-188, "at P3….G1/G0 arrested pro-spermatogonia" should be corrected. In addition, Edu labeling experiments should be performed for GFRa1+ cells or at P3 (Figure 6D-I), which may provide clues on whether cell cycle and proliferation of SSCs or pro-SGs are affected.

5. Differentially expressed genes drawn from RNA sequencing data, including scRNAseq data, should be provided as Source Tables.

6. In the Introduction, the rationale of studying SRSF10 in male germ line should be mentioned. In the Discussion, alternative implications and caveats of results should be discussed, including: what the DSSC3 mean for the mutants? What could be the reason that DEGs were found in mutants when SRSF10 is a splicing regulator and how genes affected by alternative splicing were not the same as DEGs from RNA sequencing analyses?

7. Materials and methods section requires substantial improvements by providing sufficient information on how the experiments were carried out, including details such as: hydration time for fixing testes, companies and catalogs for equipment used, antibodies (mono-/polyclonal) types, solution used for diluting antibodies, pH of PBS used, how Edu staining was done briefly, were the scRNAseq done for THY1+KIT+ and THY-/KIT+ cells, were all experiments repeated at least three times including RNA sequencing, etc.

Some other concerns include:

1. Table-1 describes the average litter size produced by wild type and mutant mice, which is higher than what is usually found for C57BL6/J mice. The number of pups per litter should be from one female mouse, please check and correct. The time period for this recording should also be indicated.

2. Change "per tubule" to "per cross-section" in Figures when applicable. The number of KIT+ and GFRa1+ cells in Figure 3D and 3F should be quantified, for example, using average number of cells/per unit length of seminiferous tubule.

3. Figure 2A, black arrows indicate apoptotic cells, what the criteria used to tell the difference? Figure 2B, when describe the comparison between control and mutants, state clearly which one is for control, which one in for mutant, lines 139-142. Figure 2E, similar to Figure 7-Supl1D, explain why the mutants contain SRSF10 protein?

4. Lines 204-207, the number of DEGs described should be the same as what are shown in Figure 4A. Line 211, "P6 specific down-regulated genes", line 218, "P8-specific down-regulated genes", if the DEGs were selected using FC > 2, they may be significantly down-regulated but not specifically.

5. Figure 5E, use the same scale on X-axis for better comparison.

6. Figure 6A, label top genes in the graph as in Figure 6-Supl2. The number of DEGs described in line 283 is not the same as what are shown in Figure 6A. Why "Cellular iron ion homeostasis" is not presented in up-regulated GOs, "Regulation of G2/M transition of mitotic cell cycle" is not in the down-regulated GOs (line 285) for USSC1 in Figure 6B?

7. Line 317-318, "we confirmed the dramatic decrease of Srsf10 in the … Srsf10cKO testes…", why is it decreased, not depleted?

8. Missing words in Figure legends, mostly after line 950; change all ">=" to correct symbol.

9. Describe what the gels shown in Figure 7E and corresponding supplements are, make sure the gel images presented are the same as what are in the source files; also for the Source Tables, check if the labeling for each Table is correct.

10. Define clearly how statistics were set for significances: *, **, ** represent what P values in Materials and methods, as well as in figure legends. Instead of "at least 500 tubules or 1000 tubules", present the exact numbers of cross-sections examined for each experiment when applicable in figure legends, e.g. for Figure 2, 3, and 6. Instead of "adults", give out the age of mice used for experiments when applicable, either in figures or figure legends.

11. Check all grammar and spelling errors throughout the manuscript.

*Reviewer #3 (Recommendations for the authors):*

I am not a developmental biologist, but find this study to be highly descriptive and without key molecular insight. Identifying the key molecular changes that are driving the loss of spermatogonia upon depletion of SRSF10 requires (1) demonstrating a change in protein expression or activity that follows the transcript changes and (2) demonstrating some or a group of these changes is sufficient to phenocopy SRSF10-depletion. While this is beyond the level of experimentation usually asked for in an *eLife* review, without this data the manuscript lacks much mechanistic impact.

---

## [Author Response]

Essential revisions:Overall, the three reviewers agree that the subject of your manuscript could be of interest to readers of eLife and that the conclusions are partly supported by the results. However, the reviewers feel that (1) the study is too descriptive and lacks molecular insights, (2) alternative explanations of the results should be considered, (3) there should be more information about whether gene expression changes are really direct targets of SRSF10, (4) there should be additional controls for several of the experiments, and (5) there should be a significant effort to correct a variety of scientific and editorial issues. Consequently, this manuscript should be extensively revised prior to further consideration for publication in eLife. The revised version of this manuscript must be submitted within 3 months of receiving this decision letter. In its present form, the manuscript is not appropriate for publication in eLife.

We thank the editor and reviewers for their constructive comments and suggestions. We have carefully addressed the concerns and revised the manuscript as suggested.

Representative comments by the reviewers follow:The molecular mechanism by which SRSF10 impacts key splicing or gene expression events, and how many of these are direct targets of SRSF10, remains unexplored. The study is highly descriptive and lacks molecular insights. Identification of the molecular changes that are driving the loss of spermatogonia upon depletion of SRSF10 requires (1) demonstrating a change in protein expression or activity that follows the transcript changes and (2) demonstrating that one or more of these changes is sufficient to phenocopy SRSF10-depletion.

We sincerely thank the reviewer for these constructive comments. We totally agree that profiling the direct targets of SRSF10 in spermatogonia is key to understanding the functional mechanism of SRSF10. To this end, we enriched the THY1^+^ spermatogonia at P6 and performed linear amplification of complementary DNA ends and sequencing (LACE-seq) ^1^(Figure 7 in the manuscript). The LACE-seq data were highly reproducible (Figure 7b) and showed that SRSF10 preferentially bound to exons and enriched between 0 and 100 nt of the 5′ and 3′ exonic sequences that flank the splice sites (Figure Ic and Id). SRSF10-binding peaks in mouse spermatogonia were significantly enriched for GA-rich (GAAGAG) consensus motifs (Figure 7d), consistent with that previously reported in chicken DT40 cells ^2^. In total, 1207 SRSF10-binding targets (FPKM≥10, fold change≥2 in both replicates) were identified in the LACE-seq data in THY1^+^ spermatogonia (Figure 7g). These genes were significantly involved in chromatin organization, cell cycle, cellular response to DNA damage stimulus critical events, stem cell population maintenance, spermatogenesis and male meiosis (Figure 7f). Combining the RNA-seq and Iso-seq data, 675 differentially spliced genes were identified, among which 113 genes (50 in RNA-seq data, *P* = 7.97e-33; 76 in Iso-seq data, *P*=1.19e-09) were directly bound by SRSF10 (Figure 8B in the manuscript). These data suggest that SRSF10 regulate alternative splicing of functional genes by directly binding to them. The remaining 562 genes with differentially spliced events might be a secondary effect of SRSF10 depletion, as SRSF10 depletion affected genes that are significantly involved in RNA splicing (Figure 8c). With respect to the differentially expressed genes, of the 206 down-regulated and 378 up-regulated genes, only 8.25% (17/260) and 1.85% (7/378) are SRSF10 targets (Figure 8d), suggesting that the changes in gene expression are unlikely a direct effect of SRSF10 depletion.

As the reviewer suggested, besides validation of the differentially spliced events at the RNA level, we performed western blot to verify the aberrant splicing at the protein level of some direct and indirect functional targets of SRSF10 in *Srsf10*-cKO spermatogonia. We used a total of 14 commercially available antibodies against 12 proteins, of which eight antibodies did not work well for western blot (Author response table 1). Four proteins with differentially spliced events were successfully validated, including two direct targets (NASP and BCLAF1) and two indirect targets (DAZL and KAT7) (Figure 8e and Figure 8E in the manuscript). Although there is no direct evidence in spermatogonia, the resulting patterns of isoform switch of NASP and BCLAF1 after Srsf10 depletion have been reported to perturb their normal function in other cell lines, especially that related to cell cycle ^3-5^, partially echoing the deficient progenitor proliferation/expansion in the absence of *Srsf10* (please also refer to lines 449-462, pages 21-22 in the manuscript).

**Author response table 1. sa2table1:** Antibodies used for validating the differentially spliced genes at the protein level in control and *Srsf10*-cKO isolated THY1^+^ spermatogonia.

Antibody	Company and Cat. No.	Work for WB	Bound by SESF10	Validated
BCLAF1	26809-1-AP Proteintech	Yes	Yes	Yes
NASP	11323-1-AP Proteintech	Yes	Yes	Yes
MCM10	12251-1-AP Proteintech	No	Yes	No
MAM10	A7199 Abclonal	Yes	Yes	No
CENPH	12841-1-AP Proteintech	Yes	Yes	No
RIF1	A15167Abclonal	No	Yes	No
KDM3B	19915-1-AP Proteintech	No	Yes	No
SYCP1	ab217295 Abcam	No	Yes	No
EXO1	16253-1-AP Proteintech	No	No	No
DAZL	MCA23336 Bio-Rad	Yes	No	Yes
RET	Ab134100 Abcam	No	No	No
KIT	AF1356R&D	No	No	No
KIT	18696-1-AP Proteintech	No	No	No
KAT7	13751-1-AP Proteintech	Yes	No	Yes

While we totally agree that it would be great if changes in expression or splicing of one or more of these genes can phenocopy that of *Srsf10* depletion, it is however sort of technically difficult to prove this. First, it is difficult to exactly mimic the changes of gene splicing, especially in vivo. Second, no spermatogonia cell lines with high transfection efficiency that can be used to test this idea are available in our hands. We believe that the phenotype of *Srsf10* depletion is more likely a combined result of abnormal splicing of many functional genes, directly or indirectly regulated by SRSF10.

Although some of the data showed that the mutant (cKO) mice may contain a deletion mutation of Srsf10 (Figure 1B, Figure 7-Supl 1D), the strategy of generating mutants and genotyping of mice should be presented as data, including the genotyping PCR (gel electrophoresis) of mice and a schematic showing the genomic organization of Srsf10 loci before and after Cre mediated recombination.

We thank the reviewer for this comment. We have added relevant information of the schematic diagram showing the deletion of *Srsf10* exon 3 and generation of *Srsf10* knockout (KO or “-”) allele by Vasa-Cre-mediated recombination in male germ cells (Figure 1A, Figure 1—figure supplement 1A in the manuscript). The breeding strategy and genotyping PCR result were also presented to show the generation of *Srsf10*^F/+^;Vasa-Cre (control) and *Srsf10*^F/-^;Vasa-Cre (*Srsf10*-cKO) mouse line (Figure 1—figure supplement 1B and 1C in the manuscript).

What does Srsf10F/- represent as described in Materials and methods (Lines 452-453)?

We apologize for not describing this clearly. In the presence of Vasa-Cre, the floxed allele in *Srsf10*^F/+^ male mouse (*Srsf10*^F/+^;Vasa-Cre) mature sperm will generate an *Srsf10* knockout (“-”) allele by Vasa-Cre-mediated recombination. In other words, in the mature sperm of *Srsf10*^F/+^;Vasa-Cre male mice, there will be no floxed allele but only the “-” allele. Thus, *Srsf10*^F/+^;Vasa-Cre males mated with *Srsf10*^F/F^ females will generate four genotypes in the progeny, including *Srsf10*^F/+^, *Srsf10*^F/-^, *Srsf10*^F/+^;Vasa-Cre and *Srsf10*^F/-^;Vasa-Cre. *Srsf10*^F/-^ represents one of the genotypes in the offspring. To avoid confusion, we have added an explanatory sentence in the Materials and methods as “*Srsf10*^F/+^;Vasa-Cre males (the floxed allele in mature sperm will be deleted by Vasa-Cre and thus generate a knockout allele (“-”)) and *Srsf10*^F/F^ females were used for breeding” (lines 490-491, page 24).

The labeling of figures can also be changed from "Control" to "Srsf10F/+; Vasa-Cre" when applicable.

We thank the reviewer for this suggestion. But if “Control” is changed to “*Srsf10*^F/+^;Vasa-Cre”, “*Srsf10*-cKO” may also need to be changed to “*Srsf10*^F/-^;Vasa-Cre”. Because the two genotypes are relatively similar (only different in “+” and “-”), it may be not so clear and readable and cause confusion to readers in the manuscript. To make this clearer, we highlighted in the manuscript as “As *Srsf10*^F/+^ and *Srsf10*^F/+^;Vasa-Cre mice were healthy and phenotypically normal, *Srsf10*^F/+^;Vasa-Cre mice (hereafter, “control”) were used as control in the following experiments unless otherwise noted.” (lines 127-128, pages 6-7).

Explain what the residual SRSF10 in Figure 7-Supl1D mean for the deletion mutation.

We thank the reviewer for this comment. The THY1^+^ spermatogonia were collected using MACS from control and *Srsf10*-cKO testes at P6. Flow cytometry showed that the proportion of THY1^+^ cells in the sorted group was 66.0%, but only 21.9% in the unsorted group. Although the proportion was greatly improved, the isolated spermatogonia were still mixed with some somatic cells in which SRSF10 was not knocked out. This might be the reason for the residual SRSF10 protein in Figure 7-Supl1D.

The authors used heterozygous mutants as the control. At least one experiment should be presented to show that the heterozygous mice indeed had same phenotypes as the wild type, and different from the homozygous mutants. This can be done by comparing spermatogenic cells on testis sections or cell sorting of SSCs among the three genotypes.

We thank the reviewer for this constructive comment. We have carefully analyzed the morphology and histology of testes from two-month-old *Srsf10*^F/+^ (wild type), *Srsf10*^F/+^;Vasa-Cre (heterozygous) and *Srsf10*^F/-^;Vasa-Cre (homozygous) males (Figure 1 in the manuscript). The weight of testes and the ratio of testes to body weight of *Srsf10*^F/+^;Vasa-Cre males were comparable to those of the *Srsf10*^F/+^ (Figure 1C-E in the manuscript). However, the weight of testes and the ratio of testes to body weight of *Srsf10*^F/-^;Vasa-Cre males were significantly reduced compared to *Srsf10*^F/+^ or *Srsf10*^F/+^;Vasa-Cre males (Figure 1C-E in the manuscript). Additionally, no obvious differences were observed in the HE-stained testis and epididymis sections of the *Srsf10*^F/+^;Vasa-Cre and *Srsf10*^F/+^ males (Figure 1F in the manuscript). Moreover, MVH^+^ and PLZF^+^ spermatogenic cells showed little difference between *Srsf10*^F/+^;Vasa-Cre and *Srsf10*^F/+^ (WT) testis sections (Figure 1G). However, much less MVH^+^PLZF^+^ spermatogonia could be observed in the center of seminiferous tubules from *Srsf10*^F/-^;Vasa-Cre mouse testes (Figure 1G). So, *Srsf10*^F/+^ and *Srsf10*^F/+^;Vasa-Cre mice were healthy and phenotypically normal. Thus, we used *Srsf10*^F/+^;Vasa-Cre mice as the control in the following experiments. We have also added relevant information in the revised manuscript (Figure 1, lines 116-132, pages 6-7)

Maintaining consistency when describing SSC sub-populations may help to avoid confusion. For example, in the Abstract it says "impeded the expansion of progenitor spermatogonia"; in lines 97-99, it "impaired … in undifferentiated spermatogonia"; in lines 234-236, it "may not affect the gene expression and proliferation of SSCs", whereas in Figure 4B, Gfra1 was among the decreased genes at P6. This may be done by clearly defining the particular cellular state with their gene expression signatures. For example, un-differentiated SSCs for cells expressing (GFRa1), undifferentiated progenitors for cells expressing OCT4 and PLZF, differentiating progenitors for cells expressing KIT, cells developed beyond type A1 stage can be referred to as differentiated spermatogonia.

We thank the reviewer for this constructive comment. This is indeed important to make the description clearer. We have added the gene expression signatures to define clearly what we were describing a particular cellular state in the manuscript as suggested. For example, we have changed “Depletion of Srsf10 in germ cells had little effect on the formation of SSCs but impeded the expansion of progenitor spermatogonia, leading to the failure of spermatogonia differentiation and meiosis initiation.” to “In the absence of SRSF10, SSCs can be formed, but the expansion of PLZF-positive undifferentiated progenitors was impaired, followed by the failure of spermatogonia differentiation (marked by KIT expression) and meiosis initiation.” in the abstract; “The cell cycle, proliferation and survival were impaired in the residual Srsf10 depleted undifferentiated spermatogonia.” to “The cell cycle, proliferation, and survival were impaired in the residual Srsf10 depleted PLZF^+^ undifferentiated progenitors.” in lines 97-98, and “Taken together, these data suggest that Srsf10 deficiency may not affect the gene expression and formation of SSCs, but disturbs genes involved in progenitor spermatogonia and thus their expansion and differentiation.” to “Taken together, these data suggest that Srsf10 deficiency disturbs genes involved in progenitor spermatogonia and thus their expansion and differentiation.” in lines 234-236 in the revised manuscript.

Other changes that can be found in the revised manuscript include lines 94-96, page 5; line 131, page 7; lines 187-189, page 9; lines 298-299, page 14; line 305, page 15.

The conclusion that "the progenitor expansion is affected, SSC formation is not" should be drawn carefully, as several of the data suggest otherwise. For example, Figure 3C, using MVH^+^PLZF- as a criterion for selecting differentiated cells may not exclude GFRa1+ SSCs, since the majority of PLZF+ progenitors are not expressing GFRa1. In this connection, Figure 3F, lines 178-179, described the decrease of GFRa1+ cells, which are classified as SSCs in the manuscript. The single cell RNAseq analyses also suggested the changes in the ratios of USSC1 (un-differentiated SSC1, increased from 17.5% to 35.5%) and USSC2 (un-differentiated SSC2, decreased from 20.2% to 8.1%) in mutants, compared to the control. Note that USSC1 expresses high GFRa1 and USSC2 expresses high PLZF and OCT4. It is likely that impeding the progenitors' cell division caused accumulation of SSCs or that SSCs' defect in M phase progression led to decrease of progenitors. This should be discussed in the Discussion section.

We thank the reviewer for this constructive comment. Indeed, the number of GFRα1^+^ cells was somewhat decreased, and the expression of *Gfrα1* was slightly down-regulated in P6 *Srsf10*-cKO testes (although most SSCs markers (*Etv5*, *Bmi1*, *Id4*, *Lhx1*, *Cd82* and *T*) was hardly affected), indicating that the formation and maintenance of GFRα1^+^ SSCs might be also affected by the depletion of SRSF10. We have toned down the conclusion in the revised manuscript (lines 28-30, page 2; lines 187-189, page 9; lines 233-236, pages 11-12).

However, the expression of SSCs markers was not up-regulated from P3 to P8, suggesting that loss of SRSF10 in spermatogonia might not cause the accumulation of SSCs. Although the ratio of USSC1 (SSCs) (now as USPG1 in the revised manuscript) was increased (from 17.5% to 35.5%) in *Srsf10*-cKO samples as detected in the scRNA-seq data, it is more likely a result of the loss of progenitors (USSC2, now as USPG2 in the revised manuscript) and differentiating cells (DSSC1 and DSSC2, now as DSPG2 and DSPG3, respectively) in the single cell suspensions subjected to scRNA-seq.

scRNA-seq data showed that the expression of genes associated with cell cycle in SSCs (USPG1) and progenitor cells (USPG2) was abnormal in *Srsf10*-cKO testes. However, EdU incorporation was reduced in PLZF^+^ spermatogonia in P6 *Srsf10*-cKO testes, but was comparable between control and *Srsf10*-cKO GFRα1^+^ cells in P3 testes, suggesting that the cell cycle defect was more severe in the progenitor cells than in SSCs. This might be due to the proliferation of progenitor spermatogonia is much faster and thus more sensitive to SRSF10 depletion than SSCs which are normally quiescent and exhibit a very slow cell cycle rate^6^. Additionally, the PLZF^+^ spermatogonia were gradually increased, but at a slower pace, in *Srsf10*-cKO testes from P3 to P12, and the expression of genes involved in the expansion and early differentiation of progenitor spermatogonia was also somewhat up-regulated from P6 to P8 *Srsf10*-cKO testes. Although we cannot fully exclude the possibility that the cell cycle defect in SSCs impedes the transition to progenitors, the significantly decreased undifferentiated spermatogonia is more likely a result of abnormal cell cycle or proliferation of progenitors.

We have revised the statement more precisely as suggested, and added a new subtitle in the Discussion section to discuss this part (lines 407-423, pages 19-20).

The authors should also introduce clearly how the single cells' classification on SSCs were done and how they would match the cell types described before RNA sequencing data.

We thank the reviewer for pointing this out. To make this clearer, we have added more information in the manuscript (lines 246-263, pages 12-13), and also in the “Data processing of scRNA-seq” part in the Materials and methods section (lines 647-659, pages 31-32). Briefly, highly variable genes across single cells were automatically identified and dimensional reduction was performed with principal components analysis (PCA) using the R package Seurat^7^. The top 16 principal components were used for cluster/subtype (five subtypes in the manuscript) identification and Uniform Manifold Approximation and Projection (UMAP) visualization^8^. Then known cell-type specific markers were used to match each cell type. Specifically, SSCs were matched to USSC1 (USPG1 in the revised manuscript), as cells in this cluster highly express SSC marker genes, including *Eomes, Id4, Lhx1, Etv5, Gfra1* and *Smoc2* (Figure 5C and Figure 5—figure supplement 1 in the manuscript). Please note that to avoid confusion, we renamed the identified five clusters from USSC to USPG, and DSSC to DSPG, respectively, in the revised manuscript.

In the Introduction, the rationale of studying SRSF10 in male germ line should be mentioned.

We thank the reviewer for this comment. We have added additional information to state the rationale of studying SRSF10 in the male germ line in the introduction (lines 82-92, pages 4-5).

In the Discussion, alternative implications and caveats of results should be discussed, including – What does DSSC3 mean for the mutants?

We thank the reviewer for this comment. The discussion has been reorganized to make the statements more precise and comprehensive. For DSSC3 (DSPG3 in the revised manuscript), they express the differentiated markers *Stra8*, *Kit* and *Sohlh1*, but at a lower level than that in the DSPG1 (DSSC1 in the previous manuscript) and DSPG2 (DSSC2 in the previous manuscript). We speculate that these cells might be directly transited from the fetal prospermatogonia in the first round of spermatogonia, or a population of abnormally differentiated cells in the process of spermatogonia differentiation, but later cell cycle defects of these cells lead to developmental arrest. Please also refer to the Discussion section in the revised manuscript (lines 437-443, page 21).

What is the reason that DEGs were found in mutants when SRSF10 is a splicing regulator?

We thank the reviewer for this interesting question. When a commonly used cutoff (FPKM ≥ 5, |log2FC| ≥ 1 and *R* < 0.05) is used, only 11 down-regulated and 60 up-regulated genes in *Srsf10*-cKO spermatogonia were identified. Of these DEGs, no genes are the direct targets of SRSF10, as detected using LACE-seq. Even if when a lower cutoff (FPKM ≥ 2, |log2FC| ≥ 0.5 and *R* < 0.05) is used, among the 206 down-regulated and 378 up-regulated genes, only 17 (8.25%) down-regulated genes and 7 (1.85%) upregulated genes were identified as the direct target of SRSF10 (Figure V and Figure 8—figure supplement 1 in the manuscript). Considering the altered cell types (for example the loss of progenitors and differentiating spermatogonia) and gene splicing in the absence of SRSF10, changes in gene expression might be mainly a secondary effect of SRSF10 depletion (lines 474-479, page 23).

How is it that genes affected by alternative splicing were not the same as DEGs from RNA sequencing analyses?

We thank the reviewer for this question. First, following the last question, we think that DEGs may be a secondary effect of SRSF10 depletion in mouse spermatogonia. Those DEGs may not be regulated directly by SRSF10, which mainly functions as a splicing factor. Second, genes that are differentially spliced do not necessarily mean that they are differentially expressed. Alternative splicing is a post-transcriptional mechanism of gene regulation, if a splicing event does not affect the stability of a transcript, but only isoform switch, the RNA level may not change.

The Materials and methods section requires substantial improvements by providing sufficient information on how the experiments were carried out, including details such as: hydration time for fixing testes, companies and catalogs for equipment used, antibodies (mono-/polyclonal) types, solution used for diluting antibodies, pH of PBS used, and how Edu staining was done. Were the scRNAseq done for THY1+KIT+ and THY-/KIT+ cells, were all experiments repeated at least three times including RNA sequencing, etc.

We appreciate these comments and suggestions. We have reorganized the Materials and methods section and provided all the suggested information in the revised manuscript.

Table-1 describes the average litter size produced by wild type and mutant mice, which is higher than what is usually found for C57BL6/J mice. The number of pups per litter should be from one female mouse, please check and correct. The time period for this recording should also be indicated.

We thank the reviewer for pointing this out and apologize for not describing this clearly. The reason why the average litter size is higher than that usually found for C57BL6/J mice is that we used ICR females for the fertility test. Additionally, the Vasa-Cre mice are on a mixed background, meaning that the *Srsf10*^F/+^;Vasa-Cre (control) and *Srsf10*^F/-^;Vasa-Cre (*Srsf10*-cKO) mice are also not pure C57B6L/J background. So, the average litter size may be higher than that for C58B6L/J mice. The time period for recording is at least 4 weeks. To make this clear, we have added the details of “Fertility test of male mice” in the Materials and methods section (lines 501-506, page 25).

Change "per tubule" to "per cross-section" in Figures when applicable. The number of KIT+ and GFRa1+ cells in Figure 3D and 3F should be quantified (e.g, using average number of cells/per unit length of seminiferous tubule).

We thank the reviewer for the suggestion. We have changed “per tubule” to “per tubule cross-section” in the Figures in the revised manuscript (Figure 1H, 3C, 3E, 3G, 3H and 6H, Figure 3—figure supplement 1B). We have quantified the number of KIT^+^ cells in Figure 3D using the number of KIT^+^ cells per 100 mm seminiferous tubule at P8 (Figure 3D in the manuscript). We have also quantified the number of GFRα1^+^ cells in Figure 3F using the number of GFRα1^+^ cells per 100 mm seminiferous tubule at P6 (Figure 3F in the manuscript).

In Figure 2A, black arrows indicate apoptotic cells. What criteria were used to tell the difference?

We thank the reviewer for the comment. Cells with round shapes, cytoplasmic and nuclear condensation in H&E staining are usually signs of apoptosis. In the seminiferous tubules of P10 and P12 testes, we observed some cells with round shapes and highly condensed cytoplasm and nuclei, which may be the degenerating apoptotic cells. As apoptosis is not the key point in this part, to avoid distraction, we have removed the description of apoptosis in the figure legend in the revised manuscript.

In Figure 2B, state clearly which one is for the control and which one in for the mutant, lines 139-142. Figure 2E is similar to Figure 7-Supl1D. Explain why the mutants contain SRSF10 protein?

We thank the reviewer for the constructive comment. We have changed the description to “Spermatocytes in the vast majority of control seminiferous tubules had entered into meiosis prophase I as expected, and abundant MVH^+^SCP3^+^ germ cells could be detected in the center. However, only very few seminiferous tubules contained a tiny minority (about 3 to 5) of MVH^+^SCP3^+^ germ cells can be observed in *Srsf10*-cKO testes.” (lines 146-149, pages 7-8). In Figure 2E, total protein from the whole testes, including both the germ cells and somatic cells, was used to examine the expression of gH2AX. SRSF10 was only specifically depleted in the germ cells. The detected residual SRSF10 protein is likely from the somatic cells in the *Srsf10*-cKO testes. In Figure7-Supl1D, the THY1^+^ spermatogonia were collected using MACS from control and *Srsf10*-cKO testes at P6. Flow cytometry showed that the proportion of THY1^+^ cells in the sorted group was 66.0%, but only 21.9% in the unsorted group. Although the proportion was greatly improved, the isolated spermatogonia were still mixed with some somatic cells in which SRSF10 was not knocked out. This might be the reason for the residual SRSF10 protein in Figure7-Supl1D.

In Figure 5E – Use the same scale on X-axis for better comparison.

We thank the reviewer for the constructive comment. We have used the same scale on X-axis in Figure 5E in the revised manuscript (Figure 5E in the revised manuscript).

In Figure 6A – Label top genes in the graph as in Figure 6-Supl2. The number of DEGs described in line 283 is not the same as that shown in Figure 6A. Why is "cellular iron ion homeostasis" not present in up-regulated Gos and "regulation of G2/M transition of mitotic cell cycle" is not present in the down-regulated GOs (line 285) for USSC1 in Figure 6B?

We thank the reviewer for pointing these out. We have labeled the top genes in the graph in the revised Figure 6A. The number of DEGs described in the previous manuscript was a careless mistake, which has been corrected in the revised manuscript as “In USPG1, 184 and 175 genes were down-regulated and up-regulated, respectively (|log2FC| > 0.25, *P*-value < 0.01)” (lines 277-278, page 13).

Genes associated with “cellular iron ion homeostasis”, including *Ftl1* and *Fth1* were indeed also up-regulated in USSC1 (now as USPG1 in the revised manuscript). However, only a related “intracellular sequestering of iron ion” is present in the GO terms, but with a higher *P*-value (0.091), which may be a result of a lack of other up-regulated genes (e.g. *Hmox1*, *Fthl17e* found in the up-regulated genes in DSSC1 and DSSC2 (now as DSPG1 and DSPG2 in the revised manuscript)) involved in this pathway. Please note that “cellular iron ion homeostasis” is also not significantly enriched in the up-regulated GO terms in USSC2 (USPG2 in the revised manuscript). We apologize for this careless mistake and have corrected it in the revised manuscript (Figure 6—figure supplement 2 A and 2B in the manuscript).

“Regulation of G2/M transition of mitotic cell cycle” is actually not present in the down-regulated GO terms in USSC1 (USPG1 in the revised manuscript) as correctly shown in Figure 6B. We described it by mistake in the previous manuscript, and have deleted this term in the revised manuscript.

Describe what the gels shown in Figure 7E and corresponding supplements are, make sure the gel images presented are the same as those in the source files, and check if the labeling for each Table is correct in the Source Files.

We thank the reviewer for pointing this out. We have added descriptions of the gel figures in the figure legends of Figure 7E and corresponding supplements.

We have checked the correspondence between each gel image and those in the source files and made sure the labeling for each Table is correct.

Define clearly how statistics were set for significances: *, **, ** represent what P values in Materials and methods, as well as in the Figure Legends. Instead of stating "at least 500 tubules or 1000 tubules", present the exact numbers of cross-sections examined for each experiment when applicable in the Figure Legends (e.g., for Figures 2, 3, and 6). Instead of "adults", give the age of mice used for experiments when applicable, either in the Figures or Figure Legends.

We thank the reviewer for this comment. We have added the relevant information in Materials and methods and in Figure Legends. “*** represents *R* < 0.001, ** represents *R* < 0.01 and * represents *R* < 0.05. *R*-value < 0.05 was considered with a significant difference level.” was added in the Statistical analysis (lines 741-742, page 36). The exact numbers of cross-section were added in the corresponding Figure Legends. We have changed the “adult” to “two-month-old mice” in the Figures and Figure Legends.

Check and correct all grammar and spelling errors throughout the manuscript.

We thank the reviewer for this comment. We have checked and corrected the grammar and spelling carefully throughout the manuscript. All the revised parts are highlighted in blue.

Reviewer #1 (Recommendations for the authors):To some extent the manuscript describes a host of molecular markers (genes) involved in alternative splicing, all of which affect steps in spermatogenesis. Consequently, it is clear that alternative splicing of pre-mRNAs is critical for spermatogenesis, but less clear as to why it is critical.

We thank the reviewer for this thought-provoking comment. This is indeed a fundamental and important question regarding how alternative splicing is involved in gene regulation and spermatogenesis. Changes in splicing would lead to changes in the expression level or the forms and functions of the protein products of a specific gene or a set of critical genes, which may further lead to the failure of spermatogenesis. In our work, we showed that depletion of *Srsf10* leads to changes in alternative splicing of functional genes in spermatogonia, including the direct targets (e.g. *Nasp* and *Bclaf1*) and the indirect targets (e.g. *Dazl* and *Kat7*). For example, isoform switches were found in *Nasp* and *Bclaf1* at both the RNA and protein levels in the absence of SRSF10. Although there is no direct evidence in spermatogonia, the resulting patterns of isoform switch of NASP and BCLAF1 after *Srsf10* depletion (Figure 8E in the manuscript) have been reported to perturb their normal function in other cell lines, especially that related to cell cycle ^3-5^, partially echoing the deficient progenitor proliferation/expansion in *Srsf10*-cKO testes (please also refer to lines 449-462, pages 21-22 in the manuscript).

It remains unclear whether or not defects in meiosis initiation by spermatogonal stem cells (SSC) in neonatal mice is due to defects in mitotic progenitor cell proliferation.

We thank the reviewer for this comment. In neonatal mice, the first round of spermatogenesis initiates directly from prospermatogonia to KIT-positive (KIT^+^) differentiating spermatogonia (thus bypassing the process through undifferentiated progenitors), which continue to develop and produce the first wave of spermatozoa. Meanwhile, another set of prospermatogonia become SSCs which are capable of self-renewal or differentiate into progenitors to initiate spermatogenesis and support the continuity of steady-state spermatogenesis. In P3 *Srsf10*-cKO testes, the number of MVH^+^ germ cells was comparable to that in control, suggesting that prospermatogonia can possibly develop into KIT^+^ differentiating spermatogonia. However, the meiosis initiation still failed in P8-P15 *Srsf10*-cKO testes during the first wave of spermatogenesis, suggesting that loss of SRSF10 might lead to developmental arrest of KIT^+^ differentiating spermatogonia, or SRSF10 might be involved in the initiation of meiosis during the first round of spermatogonia. So, the defects in meiosis initiation of the first round of spermatogenesis are not likely due to the defects in mitotic progenitor cell proliferation, because these spermatogonia do not pass through the progenitor cell proliferation stage. By contrast, the defects in meiosis initiation of subsequent rounds of spermatogenesis are likely a result of defects in mitotic progenitor cell proliferation.

Is it clear that certain steps in SSC formation are not affected by the deletion of Srsf10? In addition to experiments clearing up this issue, the authors should provide more convincing evidence that they are dealing with completely inactivated Srsf10 knockout mice (e.g., show PCR analysis etc.). Are redundant mechanisms at play here?

We thank the reviewer for this comment. The SSCs can be formed in the *Srsf10* knockout mice, but the number of GFRα1^+^ cells was somewhat decreased, and the expression of GFRα1 was slightly down-regulated in P6 *Srsf10*-cKO testes (although most SSCs markers (*Etv5*, *Bmi1*, *Id4*, *Lhx1*, *Cd82* and *T*) was hardly affected). These data indicate that the formation and maintenance of GFRα1^+^ SSCs might be also affected by the depletion of SRSF10. We have revisited this point and made revisions in the revised manuscript (lines 185-187, page 9).

The representative genotyping results for *Srsf10*-cKO mice were shown in Figure IIIc. Besides that at P8 (Figure 1B in the manuscript), we further detected the expression of SRSF10 in germ cells at P3 by co-staining SRSF10 and MVH. Compared to control, the expression of SRSF10 in germ cells was drastically reduced in *Srsf10*-cKO P3 testes (Author response image 1). We cannot exclude the possibility that redundant mechanisms exist beyond SRSF10 in SSCs formation.

**Author response image 1. sa2fig1:** Co-staining for SRSF10 and MVH in P3 control and *Srsf10*-cKO testes. White arrows indicate the representative germ cells. MVH (a germ cell marker) was co-stained to indicate the location of germ cells. The DNA was stained with Hoechst 33342. Scale bar, 20 µm.

Of the hundreds of genes in spermatogonia affected by the knockout of Srsf10 (i.e., affected by a lack of alternative splicing), which are thought to be involved in the initiation of meiosis? Are any of these same genes involved in a similar process in females, i.e., the conversion of oogonia (mitotic) into oocytes (meiotic)?

We thank the reviewer for this comment. *Stra8* (directly bound by SRSF10) and *Dazl* (not directly bound by SRSF10) are among the genes that are affected by *Srsf10* depletion and are essential for meiosis initiation^9,10^. These genes are also involved in meiosis initiation in females. However, whether depletion of *Srsf10* in females would affect the conversion of oogonia (mitotic) into oocytes (meiotic), and whether similar sets of genes are affected are unknown. Vasa-Cre mediated gene depletion happens after embryonic day 15 (E15), which is even later than the time point at which meiosis initiates in females (E13.5). For this case, a mouse model with a Cre that is expressed earlier (e.g. the Tnap-Cre) may be more suitable to investigate this question.

The manuscript should undergo major editorial improvements making it more understandable to scientists unfamiliar with spermatogenesis, Srsf10, and alternative splicing (also check the many spelling and grammatical errors).

We thank the reviewer for this comment. We have reorganized the manuscript and added additional information to make it clearer and more understandable. We have also checked and corrected the grammar and spelling errors carefully throughout the manuscript.

Reviewer #2 (Recommendations for the authors):Some main concerns that should be addressed:1. Although some of the data showed that the mutant (cKO) mice may indeed contain deletion mutation of Srsf10 (Figure 1B, Figure 7-Supl1D), the strategy of generating mutants and genotyping of mice should be presented as data, including the genotyping PCR (gel electrophoresis) of mice, a schematic showing the genomic organization of Srsf10 loci before and after Cre mediated recombination. For the same reason, what Srsf10F/- represents, as described in Materials and methods (Lines 452-453)? The labeling of figures can also be changed from "Control" to "Srsf10F/+;Vasa-Cre" when applicable. Explain what the residual SRSF10 in Figure 7-Supl1D mean for the deletion mutation.2. The authors used heterozygous mutants as the control. At least one experiment should be presented to show that the heterozygous mice indeed had same phenotypes as the wild type, and different from the homozygous mutants. This can be done by comparing spermatogenic cells on testis sections, or cell sorting of SSCs, among three genotypes.3. Maintaining consistency when describing SSC sub-populations may help to avoid confusion. For example, in Abstract, "impeded the expansion of progenitor spermatogonia"; in lines 97-99, "impaired … in undifferentiated spermatogonia"; in lines 234-236, "may not affect the gene expression and proliferation of SSCs", whereas in Figure 4B, Gfra1 was among the decreased genes at P6. It may be done by clearly defining the particular cellular state with their gene expression signatures, for example: un-differentiated SSCs for cells expressing (GFRa1), undifferentiated progenitors for cells expressing OCT4 and PLZF, differentiating progenitors for cells expressing KIT, cells developed beyond type A1 stage can be referred to as differentiated spermatogonia.4. The conclusion of "the progenitor expansion is affected, SSC formation is not" should be drawn carefully, as several data suggest otherwise. For example, Figure 3C, using MVH^+^PLZF- as a criterion for selecting differentiated cells may not exclude GFRa1+ SSCs, since the majority of PLZF+ progenitors are not expressing GFRa1. In fact, Figure 3F, lines 178-179, described the decrease of GFRa1+ cells, which are classified as SSCs in the manuscript. The single cell RNAseq analyses also suggested the changes in the ratios of USSC1 (un-differentiated SSC1, increased from 17.5% to 35.5%) and USSC2 (un-differentiated SSC2, decreased from 20.2% to 8.1%) in mutants, comparing to the control. Note that USSC1 expresses high GFRa1 and USSC2 expresses high PLZF and OCT4. It is likely that impeding progenitors' cell division caused accumulation of SSCs, or SSCs' defect in M phase progression led to decrease of progenies (progenitors). This should be discussed in the Discussion section. The authors should also introduce clearly how the single cells' classification on SSCs were done and how they would match the cell types described before RNA sequencing data.

We thank the reviewer for these valuable comments. For the above comments, please refer to the responses in Essential revision (Comments 2-9).

In the same vein, Figure 3G used MVH to label all germ cells, without considering the SSC sub-groups that may already start to differ at this time point (to support the first wave spermatogenesis), thus this data cannot say about changes in SSC formation. Here, it should be noted that pro-SGs (a.k.a. gnocytes) resume proliferation following the birth of the animal, concomitantly with their migration. The notion described in the text, lines 183-184, "they migrate … then restore proliferation", lines 186-188, "at P3….G1/G0 arrested pro-spermatogonia" should be corrected. In addition, Edu labeling experiments should be performed for GFRa1+ cells or at P3 (Figure 6D-I), which may provide clues on whether cell cycle and proliferation of SSCs or pro-SGs are affected.

We thank the reviewer for this comment. We have corrected the description of the number of MVH cells at P3 in the revised manuscript as “Because *Vasa-Cre* recombinase is expressed as early as E15.5, we examined the number of germ cells in both control and *Srsf10*-cKO testes at P3. There was no difference in germ cell numbers between control and *Srsf10*-cKO testes (Figure 3G)” (lines 190-192, pages 9-10).

As suggested, we performed EdU incorporation experiment in spermatogonia expressing GFRa1 in the control and *Srsf10*-cKO testes at P3 (Figure 6—figure supplement 3 in the manuscript). Co-staining for EdU and GFRa1 showed that EdU incorporation of GFRa1^+^ cells was comparable in control and *Srsf10*-cKO P3 testes (Figure 6—figure supplement 3 in the manuscript), suggesting that the proliferation of SSCs or pro-spermatogonia was probably not much affected at P3.

5. Differentially expressed genes drawn from RNA sequencing data, including scRNAseq data, should be provided as Source Tables.

We thank the reviewer for this comment. The differentially expressed genes of RNA-seq and scRNA-seq have been provided in the corresponding source data.

6. In the Introduction, the rationale of studying SRSF10 in male germ line should be mentioned. In the Discussion, alternative implications and caveats of results should be discussed, including: what the DSSC3 mean for the mutants? What could be the reason that DEGs were found in mutants when SRSF10 is a splicing regulator and how genes affected by alternative splicing were not the same as DEGs from RNA sequencing analyses?

We thank the reviewer for these comments. Please refer to the responses in Essential revision (Comments 10-13).

7. Materials and methods section requires substantial improvements by providing sufficient information on how the experiments were carried out, including details such as: hydration time for fixing testes, companies and catalogs for equipment used, antibodies (mono-/polyclonal) types, solution used for diluting antibodies, pH of PBS used, how Edu staining was done briefly, were the scRNAseq done for THY1+KIT+ and THY-/KIT+ cells, were all experiments repeated at least three times including RNA sequencing, etc.

We thank the reviewer for these comments. Please refer to the responses in Essential revision (Comments 14).

Some other concerns include:1. Table-1 describes the average litter size produced by wild type and mutant mice, which is higher than what is usually found for C57BL6/J mice. The number of pups per litter should be from one female mouse, please check and correct. The time period for this recording should also be indicated.2. Change "per tubule" to "per cross-section" in Figures when applicable. The number of KIT+ and GFRa1+ cells in Figure 3D and 3F should be quantified, for example, using average number of cells/per unit length of seminiferous tubule.3. Figure 2A, black arrows indicate apoptotic cells, what the criteria used to tell the difference? Figure 2B, when describe the comparison between control and mutants, state clearly which one is for control, which one in for mutant, lines 139-142. Figure 2E, similar to Figure 7-Supl1D, explain why the mutants contain SRSF10 protein?

We thank the reviewer for the above comments and suggestions. Please refer to the responses in Essential revision (Comments 15-18).

4. Lines 204-207, the number of DEGs described should be the same as what are shown in Figure 4A. Line 211, "P6 specific down-regulated genes", line 218, "P8-specific down-regulated genes", if the DEGs were selected using FC > 2, they may be significantly down-regulated but not specifically.

We thank the reviewer for this comment. The number of DEGs described in the previous manuscript was a careless mistake, which has been corrected in the revised manuscript as “However, 138 genes were differentially expressed in the *Srsf10*-cKO testes at P6 and nearly all genes (135/138) were down-regulated in *Srsf10*-cKO testes (Figure 4A).” We have changed “P6 specific down-regulated genes” and “P8-specific down-regulated genes” to “many genes that were only significantly down-regulated at P6” and “many genes that were only significantly down-regulated at P8”, respectively, in the revised manuscript (lines 215-216 and 219-220, page 11).

5. Figure 5E, use the same scale on X-axis for better comparison.6. Figure 6A, label top genes in the graph as in Figure 6-Supl2. The number of DEGs described in line 283 is not the same as what are shown in Figure 6A. Why "Cellular iron ion homeostasis" is not presented in up-regulated GOs, "Regulation of G2/M transition of mitotic cell cycle" is not in the down-regulated GOs (line 285) for USSC1 in Figure 6B?

For comments 5 and 6, please refer to the responses in Essential revision (Comments 19-20).

7. Line 317-318, "we confirmed the dramatic decrease of Srsf10 in the … Srsf10cKO testes…", why is it decreased, not depleted?

We thank the reviewer for this comment. The THY1^+^ spermatogonia were collected using MACS from control and *Srsf10*-cKO testes at P6. Flow cytometry showed that the proportion of THY1^+^ cells in the sorted group was 66.0%, but only 21.9% in the unsorted group. Although the proportion was greatly improved, the isolated spermatogonia were still mixed with some somatic cells in which SRSF10 was not knocked out. So, we described it as “decreased” instead of “depleted”.

8. Missing words in Figure legends, mostly after line 950; change all ">=" to correct symbol.

We thank the reviewer for pointing this out. The missing words mainly resulted from font issues, which have been corrected in the revised manuscript. All “>=” has been changed to “≥”.

9. Describe what the gels shown in Figure 7E and corresponding supplements are, make sure the gel images presented are the same as what are in the source files; also for the Source Tables, check if the labeling for each Table is correct.10. Define clearly how statistics were set for significances: *, **, ** represent what P values in Materials and methods, as well as in figure legends. Instead of "at least 500 tubules or 1000 tubules", present the exact numbers of cross-sections examined for each experiment when applicable in figure legends, e.g. for Figure 2, 3, and 6. Instead of "adults", give out the age of mice used for experiments when applicable, either in figures or figure legends.11. Check all grammar and spelling errors throughout the manuscript.

For comments 9-11, please refer to the responses in Essential revision (Comments 21-23).

Reviewer #3 (Recommendations for the authors):I am not a developmental biologist, but find this study to be highly descriptive and without key molecular insight. Identifying the key molecular changes that are driving the loss of spermatogonia upon depletion of SRSF10 requires (1) demonstrating a change in protein expression or activity that follows the transcript changes and (2) demonstrating some or a group of these changes is sufficient to phenocopy SRSF10-depletion. While this is beyond the level of experimentation usually asked for in an eLife review, without this data the manuscript lacks much mechanistic impact.

We thank the reviewer for these constructive and valuable comments. For these comments, please refer to the responses in Essential revision (Comment 1).

References

1 Su, R. B. *et al.* Global profiling of RNA-binding protein target sites by LACE-seq. *Nat Cell Biol* 23, 664-+, doi:10.1038/s41556-021-00696-9 (2021).

2 Zhou, X. *et al.* Transcriptome analysis of alternative splicing events regulated by SRSF10 reveals position-dependent splicing modulation. *Nucleic Acids Res* 42, 4019-4030, doi:10.1093/nar/gkt1387 (2014).

3 Alekseev, O. M., Richardson, R. T., Tsuruta, J. K. & O'Rand, M. G. Depletion of the histone chaperone tNASP inhibits proliferation and induces apoptosis in prostate cancer PC-3 cells. *Reprod Biol Endocrin* 9, doi:Artn 50 10.1186/1477-7827-9-50 (2011).

4 Fang, J. Z. *et al.* Downregulation of tNASP inhibits proliferation through regulating cell cycle-related proteins and inactive ERK/MAPK signal pathway in renal cell carcinoma cells. *Tumor Biol* 36, 5209-5214, doi:10.1007/s13277-015-3177-9 (2015).

5 Zhou, X. X. *et al.* BCLAF1 and its splicing regulator SRSF10 regulate the tumorigenic potential of colon cancer cells. *Nature Communications* 5, doi:ARTN 4581 10.1038/ncomms5581 (2014).

6 de Rooij, D. G. & Russell, L. D. All you wanted to know about spermatogonia but were afraid to ask. *J Androl* 21, 776-798 (2000).

7 Satija, R., Farrell, J. A., Gennert, D., Schier, A. F. & Regev, A. Spatial reconstruction of single-cell gene expression data. *Nat Biotechnol* 33, 495-502, doi:10.1038/nbt.3192 (2015).

8 Becht, E. *et al.* Dimensionality reduction for visualizing single-cell data using UMAP. *Nat Biotechnol*, doi:10.1038/nbt.4314 (2018).

9 Anderson, E. L. *et al.* Stra8 and its inducer, retinoic acid, regulate meiotic initiation in both spermatogenesis and oogenesis in mice. *P Natl Acad Sci USA* 105, 14976-14980, doi:10.1073/pnas.0807297105 (2008).

10 Lin, Y. F., Gill, M. E., Koubova, J. & Page, D. C. Germ Cell-Intrinsic and -Extrinsic Factors Govern Meiotic Initiation in Mouse Embryos. *Science* 322, 1685-1687, doi:10.1126/science.1166340 (2008).